# Training on Foveated Images Improves Robustness to Adversarial Attacks

**Muhammad A. Shah**
Language Technologies Institute
Carnegie Mellon University
Pittsburgh, PA 15213
mshah1@cmu.edu

**Aqsa Kashaf** [*]
ByteDance
San Jose, CA 95110
akashaf@cmu.edu

**Bhiksha Raj**
Language Technologies Institute
Carnegie Mellon University
Pittsburgh, PA 15213
bhiksha@cs.cmu.edu

## Abstract

Deep neural networks (DNNs) are known to be vulnerable to adversarial attacks – subtle, perceptually indistinguishable perturbations of inputs that change the response of the model. We hypothesize that an important contributor to the robustness of human visual perception is constant exposure to low-fidelity visual stimuli in our peripheral vision. To investigate this hypothesis, we develop *R-Blur*, an image transform that simulates the loss in fidelity of peripheral vision by blurring the image and reducing its color saturation based on the distance from a given fixation point. We show that compared to DNNs trained on the original images, DNNs trained on images transformed by *R-Blur* are substantially more robust to adversarial attacks, as well as other, non-adversarial, corruptions, achieving up to 25% higher accuracy on perturbed data[2].

## 1 Introduction

Deep Neural Networks (DNNs) are exceptionally adept at many computer vision tasks and have emerged as one of the best models of the biological neurons involved in visual object recognition [1, 2]. However, their lack of robustness to subtle image perturbations that humans are largely invariant [3, 4, 5] to has raised questions about their reliability in real-world scenarios. Of these perturbations, perhaps the most alarming are *adversarial attacks*, which are specially crafted distortions that can change the response of DNNs when added to their inputs [3, 6] but are either imperceptible to humans or perceptually irrelevant enough to be ignored by them.

While several defenses have been proposed over the years to defend DNNs against adversarial attacks, only a few of them have sought inspiration from biological perception, which, perhaps axiomatically, is one of the most robust perceptual systems in existence. Instead, most methods seek to *teach* DNNs to be robust to adversarial attacks by exposing them to adversarially perturbed images [7, 8, 9] or random noise [10, 11, 12] during training. While this approach is highly effective in making DNNs robust to the types of perturbations used during training, the robustness often does not generalize to other types of perturbations [13, 14, 15]. In contrast, biologically-inspired defenses seek to make DNNs robust by integrating into them biological mechanisms that would bring their behavior more in line with human/animal vision [16, 17, 18, 19, 20, 21, 22]. As these defenses do not require DNNs to be trained on any particular type of perturbation, they yield models that, like humans, are robust to a variety of perturbations [18] in addition to adversarial attacks. For this reason, and in light of the evidence indicating a positive correlation between biological alignment and adversarial robustness [18, 23], we believe biologically inspired defenses are more promising in the long run.

---

[*]work done while at Carnegie Mellon University

[2]The code for *R-Blur* is available at https://github.com/ahmedshah1494/RBlur

37th Conference on Neural Information Processing Systems (NeurIPS 2023).

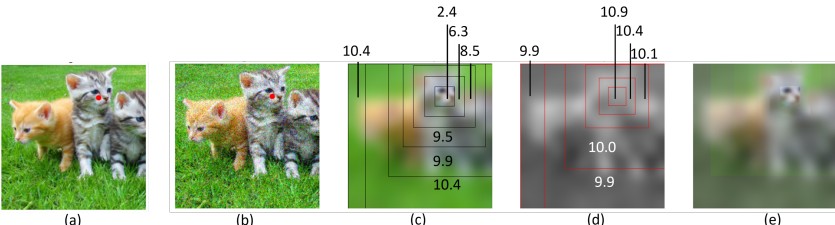

Figure 1: *R-Blur* adds Gaussian noise to image (a) with the fixation point (red dot) to obtain (b). It then creates a colored and a grayscaled copy of the image and applies adaptive Gaussian blurring to them to obtain the low-fidelity images (c) and (d), where the numbers indicate the standard deviation of the Gaussian kernel applied in the region bounded by the boxes. The blurred color and gray images are combined in a pixel-wise weighted combination to obtain the final image (e), where the weights of the colored and gray pixels are a function of their respective estimated acuity values (see 2.2).

Following this line of inquiry, we investigate the contribution of low-fidelity visual sensing that occurs in peripheral vision to the robustness of human/animal vision. Unlike DNNs, which sense visual stimuli at maximum fidelity at every point in their visual field, humans sense most of their visual field in low fidelity, i.e without fine-grained contrast [24] and color information [25]. In adults with fully developed vision, only a small region (less than 1% by area) of the visual field around the point of fixation [26] can be sensed with high fidelity. In the remainder of the visual field (the periphery), the fidelity of the sensed stimuli decreases exponentially with distance from the fixation point [27]. This phenomenon is called "foveation". Despite this limitation, humans can accurately categorize objects that appear in the visual periphery into high-level classes [28]. Meanwhile, the presence of a small amount of noise or blurring can decimate the accuracy of an otherwise accurate DNN. Therefore, we hypothesize that the experience of viewing the world at multiple levels of fidelity, perhaps even at the same instant, causes human vision to be invariant to low-level features, such as textures, and high-frequency patterns, that can be exploited by adversarial attacks.

In this paper, we propose *R-Blur* (short for Retina Blur), which simulates foveation by blurring the image and reducing its color saturation adaptively based on the distance from a given fixation point. This causes regions further away from the fixation point to appear more blurry and less vividly colored than those closer to it. Although adaptive blurring methods have been proposed as computational approximations of foveation [29, 30, 31], their impact on robustness has not been evaluated to the best of our knowledge. Furthermore, color sensitivity is known to decrease in the periphery of the visual field [25, 32, 33], yet most of the existing techniques do not account for this phenomenon.

Similar to how the retina preprocesses the visual stimuli before it reaches the visual cortex, we use *R-Blur* to preprocess the input before it reaches the DNN. To measure the impact of *R-Blur*, we evaluate the object recognition capability of ResNets [34] trained with and without *R-Blur* on three image datasets: CIFAR-10 [35], Ecoset [36] and Imagenet [37], under different levels of adversarial attacks and common image corruptions [38]. We find that *R-Blur* models retain most of the high classification accuracy of the base ResNet while being more robust. Compared to the base ResNet, *R-Blur* models achieve 12-25 percentage points (pp) higher accuracy on perturbed images. Furthermore, the robustness achieved by *R-Blur* is certifiable using the approach from [10]. We also compare *R-Blur* with two biologically inspired preprocessing defenses, namely *VOneBlock* [18], a fixed parameter module that simulates the primate V1, and a non-uniform sampling-based foveation technique [22], which we refer to as *R-Warp*. We observe that *R-Blur* induces a higher level of robustness, achieving accuracy up to 33 pp higher than *R-Warp* and up to 15 pp higher than *VOneBlock* against adversarial attacks. Compared to adversarial training (*AT*) [7, 8] – the state-of-the-art non-biological defense, *R-Blur* achieves up to 7 pp higher accuracy on average against non-adversarial corruptions of various types and strengths thus indicating that the robustness of *R-Blur* generalizes better to non-adversarial perturbations than *AT*. Finally, an ablation study showed that both adaptive blurring and desaturation contribute to the improved robustness of *R-Blur*.

## 2   Retinal Blur: An Approximation for Peripheral Vision

To simulate the loss in contrast and color sensitivity of human perception with increasing eccentricity, we propose *R-Blur*, an adaptive Gaussian blurring, and color desaturation technique. The operations

performed by *R-Blur*, given an image and fixation point, are shown in Figure 1. First, *R-Blur* adds Gaussian noise to the image to simulate stochastic firing rates of biological photoreceptors [39]. It then creates color and grayscale copies of the image and estimates the acuity of color and grayscale vision at each pixel location, using distributions that approximate the relationship between distance from the fixation point (eccentricity) and visual acuity levels in humans. *R-Blur* then applies *adaptive* Gaussian blurring to both image copies such that the standard deviation of the Gaussian kernel at each pixel in the color and the grayscale image is a function of the estimated color and grayscale acuity at that pixel. Finally, *R-Blur* combines the two blurred images in a pixel-wise weighted combination in which the weights of the colored and gray pixels are a function of their respective estimated acuity values. Below we describe some of the more involved operations in detail.

## 2.1 Eccentricity Computation

The distance of a pixel location from the fixation point, i.e. its eccentricity, determines the standard deviation of the Gaussian kernel applied to it and the combination weight of the color and gray images at this location. While eccentricity is typically measured radially, in this paper we use a different distance metric that produces un-rotated square level sets. This property allows us to efficiently extract regions having the same eccentricity by simply slicing the image tensor. Concretely, we compute the eccentricity of the pixel at location $(x_p, y_p)$ as

$$e_{x_p, y_p} = \frac{\max(|x_p - x_f|, |y_p - y_f|)}{W_V},\tag{1}$$

where $(x_f, y_f)$ and $W_V$ represent the fixation point and the width of the visual field, i.e. the rectangular region over which *R-Blur* operates and defines the maximum image size that is expected by *R-Blur*. We normalize by $W_V$ to make the $e_{x_p, y_p}$ invariant to the size of the visual field.

## 2.2 Visual Acuity Estimation

We compute the visual acuity at each pixel location based on its eccentricity. The biological retina contains two types of photoreceptors. The first type, called cones, are color sensitive and give rise to high-fidelity visual perception at the fovea, while the second type, called rods, are sensitive to only illumination but not color and give rise to low-fidelity vision in the periphery. We devise the following two sampling distributions, $D_R(e_{x,y})$ and $D_C(e_{x,y})$, to model the acuity of color and grayscale vision, arising from the cones and rods at each pixel location, $(x, y)$.

$$\mathcal{D}(e; \sigma, \alpha) = \max\left[\lambda(e; 0, \sigma), \gamma(e; 0, \alpha\sigma)\right]\tag{2}$$
$$D_C(e; \sigma_C, \alpha) = \mathcal{D}(e; \sigma_C, \alpha)\tag{3}$$
$$D_R(e; \sigma_R, \alpha, p_{max}) = p_{max}(1 - \mathcal{D}(e; \sigma_R, \alpha)),\tag{4}$$

where $\lambda(.; \mu, \sigma)$ and $\gamma(.; \mu, \sigma)$ are the PDFs of the Laplace and Cauchy distribution with location and scale parameters $\mu$ and $\sigma$, and $\alpha$ is a parameter used to control the width of the distribution. We set $\sigma_C = 0.12, \sigma_R = 0.09, \alpha = 2.5$ and $p_{max} = 0.12$. We choose the above equations and their parameters to approximate the curves of photopic and scotopic visual acuity from [27]. The resulting acuity estimates are shown in Figure 2b. Unfortunately, the measured photopic and scotopic acuity curves from [27] cannot be reproduced here due to copyright reasons, however, they can be viewed at https://nba.uth.tmc.edu/neuroscience/m/s2/chapter14.html (see Figure 14.3).

## 2.3 Quantizing the Visual Acuity Estimate

In the form stated above, we would need to create and apply as many Gaussian kernels as the distance between the fixation point and the farthest vertex of the visual field. This number can be quite large as the size of the image increases and will drastically increase the per-image computation time. To mitigate this issue we quantize the estimated acuity values. As a result, the locations to which the same kernel is applied no longer constitute a single pixel perimeter but become a much wider region (see Figure 1 (c) and (d)), which allows us to apply the Gaussian kernel in these regions very efficiently using optimized implementations of the convolution operator.

To create a quantized eccentricity-acuity mapping, we do the following. We first list all the color and gray acuity values possible in the visual field by assuming a fixation point at $(0, 0)$, computing eccentricity values $e_{0,y}$ for $y \in [0, W_V]$ and the corresponding values of $\mathcal{D}_R = \{D_R(e_{0,y}) | y \in$

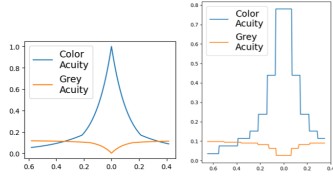
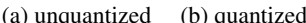

(a) unquantized  (b) quantized

Figure 2: Estimated visual acuity of sharp and colorful, photopic, and gray and blurry, scotopic, vision using equations 3 and 4

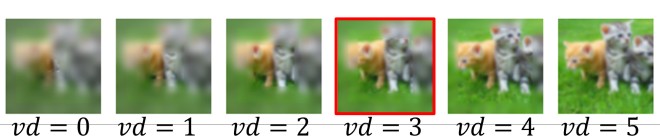

$vd = 0 \quad vd = 1 \quad vd = 2 \quad vd = 3 \quad vd = 4 \quad vd = 5$

Figure 3: Illustration of increasing the viewing distance (left to right). As the viewing distance is increased, more of the image is brought into focus. We used $vd = 3$ during inference.

$[0, W_V]\}$ and $\mathcal{D}_C = \{D_C(e_{0,y})|y \in [0, W_V]\}$. We then compute and store the histograms, $H_R$ and $H_C$, from $\mathcal{D}_R$ and $\mathcal{D}_C$, respectively. To further reduce the number of kernels we need to apply and increase the size of the region each of them is applied to, we merge the bins containing less than $\tau$ elements in each histogram with the adjacent bin to their left. After that, given an image to process, we will compute the color and gray visual acuity for each pixel, determine in which bin it falls in $H_R$ and $H_C$, and assign it the average value of that bin.

## 2.4 Changing the Viewing Distance

Increasing the viewing distance can be beneficial as it allows the viewer to gather a more global view of the visual scene and facilitates object recognition. To increase the viewing distance we drop the $k$ lowest acuity bins and shift the pixels assigned to them $k$ bins ahead such that the pixels that were in bins 1 through $k - 1$ are now assigned to bin 1. Figure 3 shows the change in the viewing distance as the value of $k$ increases from 0 to 5. Formally, given the quantized $D_C(e_{x,y})$ and $D_R(e_{x,y})$, let $D = [d_1, ..., d_n]$ represent the value assigned to each bin and $P_i$ be the pixel locations assigned to the $i^{th}$ bin, with $P_1$ and $P_n$ corresponding to points with the lowest and highest eccentricity, respectively. To increase the viewing distance, we merge bins 1 through $k$ such that $D' = [d_1, ..., d_{n-k}]$ and the corresponding pixels are $P_1' = [P_1, ..., P_k]$ and $P_{i>1} = P_{k+1}$.

## 2.5 Blurring and Color Desaturation

We map the estimated visual acuity at each pixel location, $(x_p, y_p)$, to the standard deviation of the Gaussian kernel that will be applied at that location as $\sigma_{(x_p, y_p)} = \beta W_V (1 - D(e_{x,y}))$, where $\beta$ is constant to control the standard deviation and is set to $\beta = 0.05$ in this paper, and $D = D_C$ for pixels in the colored image and $D = D_R$ for pixels in the grayscaled image. We then apply Gaussian kernels of the corresponding standard deviation to each pixel in the colored and grayscale image to obtain an adaptively blurred copy of each, which we combine in a pixel-wise weighted combination to obtain the final image. The weight of each colored and gray pixel is given by the normalized color and gray acuity, respectively, at that pixel. Formally, the pixel at $(x_p, y_p)$ in the final image has value

$$v_{(x_p,y_p)}^f = \frac{v_{(x_p,y_p)}^c D_C(e_{x,y}; \sigma_C, \alpha) + v_{(x_p,y_p)}^g D_R(e_{x,y}; \sigma_C, \alpha)}{D_C(e_{x,y}; \sigma_C, \alpha) + D_R(e_{x,y}; \sigma_C, \alpha)}, \tag{5}$$

$v_{(x_p,y_p)}^c$ and $v_{(x_p,y_p)}^g$ are the pixel value at $(x_p, y_p)$ in the blurred color and gray images respectively.

## 3 Evaluation

In this section, we determine the accuracy and robustness of *R-Blur* by evaluating it on clean data and data that has been perturbed by either adversarial attacks or common – non-adversarial – corruptions. We compare the performance of *R-Blur* with an unmodified ResNet, two existing biologically-inspired defenses, *R-Warp* [22] and *VOneBlock* [18], and a non-biological adversarial defenses: Adversarial Training (*AT*) [7]. We show that *R-Blur* is significantly more robust to adversarial attacks and common corruptions than the unmodified ResNet and prior biologically inspired methods. Moreover, we use Randomize Smoothing [10] to show that *R-Blur* is *provably* robust. While *AT* is more robust

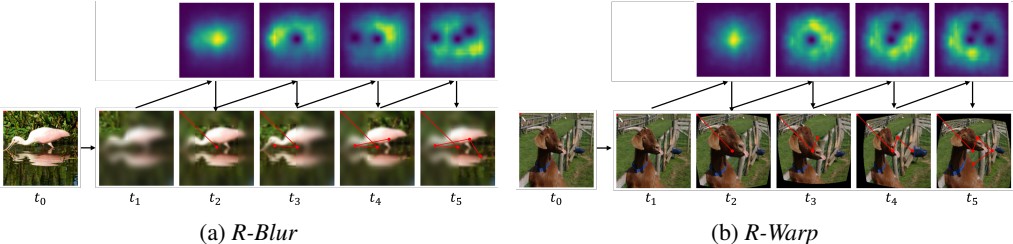

|(a) *R-Blur*|(b) *R-Warp*|

Figure 4: Illustration of fixation selection. The initial fixation point is set to top-left (0,0) and the image at $t_0$ is processed with *R-Blur* /*R-Warp* to get the image at $t_1$. DeepGaze-III is used to generate a fixation heatmap from this image. The next fixation point is sampled from the heat map, and *R-Blur* /*R-Warp* is applied to get the image at $t_2$. The region in the heatmap around the chosen fixation point is masked with an inverted Gaussian kernel to prevent spatial clustering of fixation points. This process is repeated to get a sequence of fixation points.

than *R-Blur* against adversarial attacks, *R-Blur* is more robust than *AT* against common corruptions, thus indicating that the robustness of *R-Blur* generalizes better to different types of perturbation than *AT*. We also analyze the contribution of the various components of *R-Blur* in improving robustness.

### 3.1 Experimental Setup

**Datasets:** We use natural image datasets, namely CIFAR-10 [35], Imagenet ILSVRC 2012 [37], Ecoset [36] and a 10-class subset of Ecoset (Ecoset-10). Ecoset contains around 1.4M images, mostly obtained from ImageNet database [40] (not the ILSVRC dataset), that are organized into 565 basic object classes. The classes in Ecoset correspond to commonly used nouns that refer to concrete objects. To create Ecoset-10, we select 10 classes from Ecoset that have the highest number of images. The training/validation/test splits of Ecoset-10 and Ecoset are 48K/859/1K, and 1.4M/28K/28K respectively. For most experiments with Ecoset and Imagenet, we use 1130, and 2000 test images, with an equal number of images per class. During training, we use random horizontal flipping and padding + random cropping, as well as AutoAugment [41] for CIFAR-10 and RandAugment for Ecoset and Imagenet. All Ecoset and Imagenet images were resized and cropped to $224 \times 224$. We applied these augmentations to *all* the models we trained – those with biological and non-biological defenses, as well as the baseline models.

**Model Architectures:** For CIFAR-10 we use a Wide-Resnet [42] model with 22 convolutional layers and a widening factor of 4, and for Ecoset and Imagenet we use XResNet-18 from fastai [43] with a widening factor of 2. Moving forward, we will refer to both these models as ResNet and indicate only the training/evaluation datasets from which the exact architecture may be inferred. Results for additional architectures are presented in Appendix C.

**Baselines and Existing Methods:** We compare the performance of *R-Blur* to two baselines: (1) an unmodified ResNet trained on clean data (ResNet), and (2) a ResNet which applies five affine transformations [3] to the input image and averages the logits (*RandAffine*). We also compare *R-Blur* with two biologically inspired defenses: *VOneBlock* pre-processing proposed in [18], which simulates the receptive fields and activations of the primate V1 [4], and *R-Warp* preprocessing proposed in [22], which simulates foveation by resampling input images such that the sampling density of pixels is maximal at the point of fixation and decays progressively in regions further away from it. Finally, we compare *R-Blur* with two non-biological adversarial defenses: fast adversarial training [8] with $\|\delta\|_\infty = 0.008$ (*AT*), and Randomized Smoothing (*RS*) [10].

**Fixation Selection for *R-Blur* and *R-Warp*:** While training models with *R-Blur* and *R-Warp*, we split each batch into sub-batches of 32 images, and for each sub-batch, we randomly sample a single fixation point that we use to apply *R-Blur* or *R-Warp* to all the images in that sub-batch. While training the *R-Blur* model, we also set the viewing distance uniformly at random using the procedure

---

[3]We apply rotation, translation, and shearing, with their parameters sampled from $[-8.6°, 8.6°]$, $[-49, 49]$ and $[-8.6°, 8.6°]$ respectively. The ranges are chosen to match the ranges used in RandAugment. The random seed is fixed during evaluation to prevent interference with adversarial attack generation.

[4]As in [18], we remove the first conv, batch norm, ReLU, and MaxPool from the ResNet with *VOneBlock*.

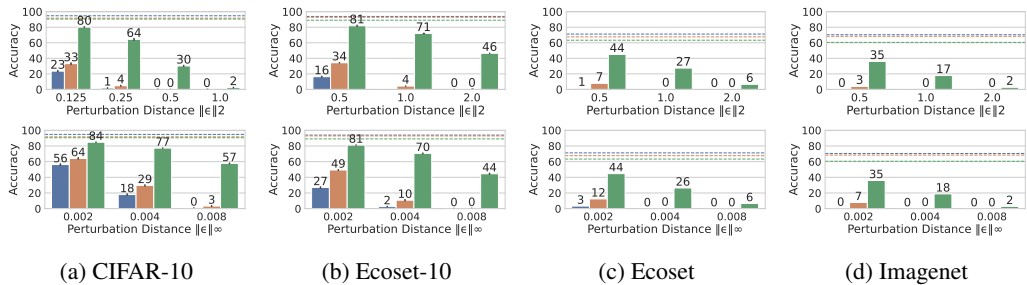

| (a) CIFAR-10 | (b) Ecoset-10 | (c) Ecoset | (d) Imagenet |

Figure 5: Comparison of accuracy on various datasets (a-d) under adversarial attacks of several $\ell_2$ (top) and $\ell_\infty$ (bottom) norms between *R-Blur* (green) and two baseline methods: *RandAffine* (orange) and ResNet (blue). The dashed lines indicate accuracy on clean images. *R-Blur* models consistently achieve higher accuracy than baseline methods on all datasets, and adversarial perturbation sizes.

described in 2.4. During inference, we determine a sequence of five fixation points (a scanpath) using DeepGaze-III [44]. DeepGaze-III passes the input image through a pretrained CNN backbone (DenseNet-201 in [44]) and extracts the activations from several intermediate layers of the CNN. It then applies a sequence of pointwise convolution and normalization layers to the activations to obtain a heatmap indicating where a human is likely to fixate. We found that it was more efficient to not use the scanpath prediction module in DeepGaze-III, and instead obtain scanpaths by keeping track of the past fixation points, and masking the predicted heatmap at these locations prior to sampling the next fixation point from it. This process is illustrated in Figure 4.

We trained two instances of DeepGaze-III using the ResNets we trained with *R-Blur* and *R-Warp* as the CNN backbone. We use the corresponding DeepGaze-III models to predict the scanpaths for *R-Blur* and *R-Warp* models. To train deepgaze-iii we used the code from the official github repository [41]. The only significant modification we made was to replace the pretrained DenseNet-201 with the pretrained R-Warp/R-Blur augmented XResNet-18 we trained on ImageNet. This improves performance, while keeping the total number of parameters low. Following [41] we train DeepGaze on the SALICON dataset [45]. This corresponds to phase 1 of training mentioned in Table 1 of [41]. We did not notice any benefits in our use case of phases 2-4, so we skipped them.

### 3.2 Results

***R-Blur* improves robustness to white-box attacks.** We evaluate robustness by measuring the accuracy of models under Auto-PGD (APGD)[46] attack, which is a state-of-the-art white-box adversarial attack. We run APGD for 25 steps on each image. We find that increasing the number of steps beyond 25 only minimally reduces accuracy (Appendix A). We take a number of measures to avoid the pitfalls of gradient obfuscation [47, 48] so that our results reflect the true robustness of *R-Blur*. These steps and detailed settings used for adversarial attacks are mentioned in Appendix A.

To determine if *R-Blur* improves robustness, we compare *R-Blur* with the unmodified ResNet and *RandAffine* under the APGD attack. We observe that *R-Blur* is significantly more robust than the unmodified ResNet and *RandAffine* models, consistently achieving higher accuracy than the two on all datasets and against all perturbation types and sizes, while largely retaining accuracy on clean data (Figure 5). While *RandAffine* does induce some level of robustness, it significantly underperforms *R-Blur*. On smaller datasets, *R-Blur* suffers relatively little loss in accuracy at small to moderate levels ($\|\delta\|_\infty \leq 0.004$, $\|\delta\|_2 \leq 1$) of adversarial perturbations, while the accuracy of baseline methods quickly deteriorates to chance or worse. On larger datasets – Ecoset and Imagenet, even the smallest amount of adversarial perturbation ($\|\delta\|_\infty = 0.002$, $\|\delta\|_2 = 0.5$) is enough to drive the accuracy of the baselines to ~10%, while *R-Blur* still is able to achieve 35-44% accuracy. As the perturbation is increased to $\|\delta\|_\infty = 0.004$ and $\|\delta\|_2 = 1.0$, the accuracy of the baselines goes to 0%, while *R-Blur* achieves 18-22%. We do observe that the accuracy of *R-Blur* on clean data from Ecoset and Imagenet is noticeably lower than that of the baseline methods.

We also compare *R-Blur* to two existing biologically motivated adversarial defenses: *VOneBlock* and *R-Warp*, and find that *R-Blur* achieves higher accuracy than both of them at all perturbation sizes and

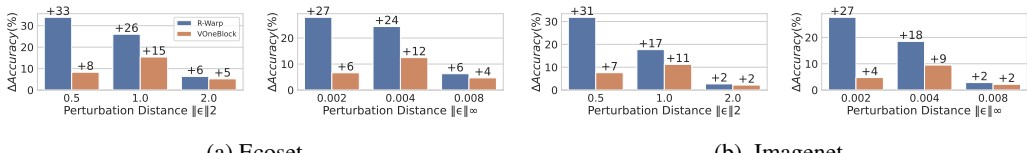

(a) Ecoset                                        (b) Imagenet

Figure 6: The difference in accuracy under adversarial attacks of several $\ell_2$ and $\ell_\infty$ norms between *R-Blur* and two biologically inspired defenses: *R-Warp* (blue) and *VOneBlock* (orange). *R-Blur* consistently achieves higher accuracy on all adversarial perturbation sizes than *R-Warp* and *VOneBlock*.

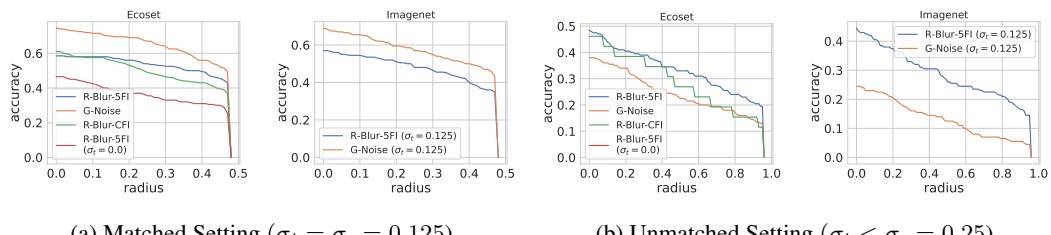

(a) Matched Setting ($\sigma_t = \sigma_c = 0.125$)          (b) Unmatched Setting ($\sigma_t < \sigma_c = 0.25$)

Figure 7: The certified accuracy at various $\ell_2$-norm radii of *R-Blur* and *G-Noise* models. *R-Blur*-CFI uses 1 fixation at the center of the image, and *R-Blur*-5FI, averages logits from 5 fixation (corners + center). $\sigma_t$ denotes the scale of noise added during training and is 0.125 unless specified, whereas $\sigma_c$ is the scale of the noise used to compute the certified accuracy. *G-Noise* outperforms *R-Blur* in the matched scenario, while *R-Blur* is superior in the unmatched scenario indicating that the robustness of *R-Blur* is more generalizable.

types. From Figure 6 we see that *R-Blur* achieves up to 33pp higher accuracy than *R-Warp*, and up to 15 pp higher accuracy than *VOneBlock* on adversarially perturbed data.

***R-Blur* is certifiably robust.** To verify that the gains in robustness observed above are indeed reliable, we use the certification method (CERTIFY) from [10] to provide formal robustness guarantees for *R-Blur*. This entails obtaining predictions for an input under a *very* large number ($10^5$) of noise samples drawn from $\mathcal{N}(0, \sigma_c)$, and using a hypothesis test to determine the *certified radius* around the input in which the model's prediction is stable *with high probability* ($\geq 99.9\%$). Given a dataset, we can compute the *certified accuracy* at a radius $r$ as the proportion of data points for which the certified radius is $\geq r$ and the model's prediction is correct. It was shown in [10] that a model trained on data perturbed with Gaussian noise achieves high certified accuracy. We call this model *G-Noise*. We compare the certified accuracy of *G-Noise* and *R-Blur* on 200 images from Imagenet and Ecoset.

We expose both *R-Blur* and *G-Noise* to Gaussian noise of scale $\sigma_t = 0.125$ during training and compute their certified accuracy at radii $r \in \{0.5, 1.0\}$. According to [10], if the scale of the noise used in CERTIFY is $\sigma_c$, then the maximum radius for which certified accuracy can be computed (with $10^5$ noise samples) is $r = 4\sigma_c$. Therefore, when computing certified accuracy at $r \leq 0.5$ CERTIFY adds noise of the same scale as was used during training ($\sigma_c = 0.125 = \sigma_t$), thus we call this the *matched* setting. However, to compute certified accuracy at $r \leq 1.0$ CERTIFY adds noise of a larger scale than was used during training ($\sigma_c = 0.25 > \sigma_t$), and thus in order to achieve high certified accuracy at $r \leq 1.0$ the model must be able to generalize to a change in noise distribution. We call this the *unmatched* setting.

Figure 7a and 7b show the certified accuracy of *R-Blur* and the *G-Noise* on Ecoset and Imagenet at several $\ell_2$ norm radii under matched and unmatched settings. In both settings, we see that *R-Blur* achieves a high certified accuracy on both Ecoset and Imagenet, with the certified accuracy at $r \approx 0.5$ and $r \approx 1.0$ being close to the ones observed in Figure 5, indicating that our earlier results are a faithful representation of *R-Blur*'s robustness. Furthermore, we see that even if *R-Blur* was trained without any noise, it can still achieve more than 50% of the certified accuracy achieved by *R-Blur* trained with noise. This indicates that adaptive blurring and desaturation do in fact endow the model with a significant level of robustness. Finally, we note that while *G-Noise* has (slightly) higher certified accuracy than *R-Blur* in the matched setting, *R-Blur* achieves significantly higher certified accuracy in the unmatched setting, outstripping *G-Noise* by more than 10 pp at $r \approx 1.0$ on Imagenet. This shows that the robustness of *R-Blur* generalizes beyond the training conditions, while *G-Noise*

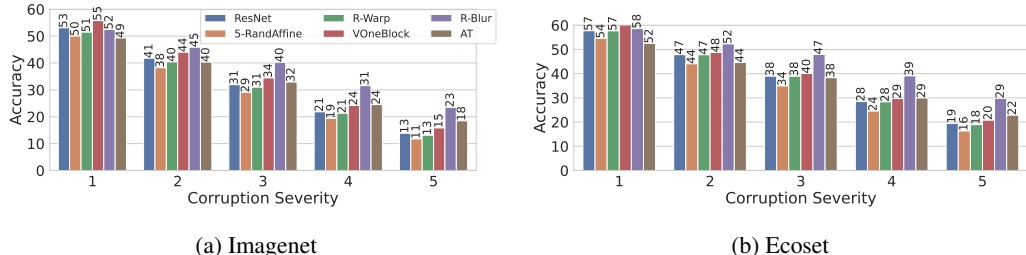

|  | (a) Imagenet | | | (b) Ecoset |
|---|---|---|---|---|

Figure 8: The accuracy of the models on Imagenet and Ecoset under the common corruptions from [38] at various severity levels. We see that *R-Blur* generally achieves the highest accuracy.

| Method | Mean | CC | WB | Clean | Mean | CC | Wb | Clean |
|---|---|---|---|---|---|---|---|---|
| | | | Ecoset | | | | Imagenet | |
| ResNet | 37.1 | 39.4 | 0.8 | 71.2 | 34.7 | 33.6 | 0.1 | **70.3** |
| *RandAffine* | 35.7 | 35.8 | 3.6 | 67.6 | 33.7 | 30.8 | 2.0 | 68.3 |
| *AT* | **49.0** | 38.5 | **47.5** | 61.1 | **46.3** | 34.2 | **43.5** | 61.3 |
| *R-Warp* | 38.5 | 40.0 | 4.5 | 71.1 | 34.1 | 32.5 | 2.2 | 67.7 |
| *VOneBlock* | 42.9 | 40.7 | 16.1 | **72.0** | 38.8 | 35.8 | 11.9 | 68.7 |
| *R-Blur* | 44.2 | **45.6** | 23.8 | 63.3 | 38.9 | **39.0** | 17.2 | 60.5 |

**best**, second best

Table 1: Accuracy of the evaluated models on clean, and perturbed data from Imagenet. "WB" refers to the accuracy under APGD attacks, while "CC" refers to the accuracy under common non-adversarial corruption [38]. *R-Blur* significantly improves the robustness of ResNet, and outperforms prior biologically motivated defenses, while approaching the performance of *AT*.

overfits to them. This makes *R-Blur* particularly suited for settings in which the exact adversarial attack budget is not known, and the model must be able to generalize.

***R-Blur* Improves accuracy on common (non-adversarial) corruptions.** Adversarial perturbations constitute only a small subset of perturbations that human vision is invariant to, therefore we evaluate *R-Blur* on a set of common image corruptions [38] that humans are largely invariant to but DNNs are not. We sample 2 images/class from Imagenet and 5 images/class from Ecoset. Then we apply 17 [5] common corruptions proposed in [38] at 5 different severity levels to generate 85 corrupted versions of each image. This yields corrupted versions of Imagenet and Ecoset containing 170K and 240K images, respectively.

Figure 8 shows the accuracy of the models on corrupted Ecoset and Imagenet. Here we also compare against an adversarially trained model (*AT*) trained with $\|\delta\|_\infty = 0.008$ using the method of [8]. We see that at severity greater than 1 *R-Blur* consistently achieves the highest accuracy. Furthermore, we also note that *R-Blur*, and *VOneBlock* consistently achieve higher accuracy than *AT*, which supports our hypothesis that the robustness of biologically motivated methods, and particularly *R-Blur*, is more general than non-biological defenses, like *AT*. In fact, the accuracy of *AT* on common corruptions is generally lesser than or at par with the accuracy of the unmodified ResNet, indicating that the robustness of *AT* does not generalize well.

**Summary of Results:** Table 1 summarizes the results of our paper and reiterates two key observations from earlier sections. Firstly, *R-Blur* makes models more significantly robust to adversarial perturbations than the unmodified ResNet, and other biologically inspired defenses. *R-Blur*, however, achieves lower accuracy against white-box attacks than *AT*. This is to be expected because *AT* is trained on adversarially perturbed data. Secondly, *R-Blur* augmented models are significantly more robust to common corruptions than all other models, including *AT*. In contrast, the accuracy of *AT* on common corruptions is almost the same as that of the unmodified ResNet, indicating that the robustness of *AT* does not generalize.

---

[5]We exclude Gaussian blur and Gaussian noise since they are similar to the transformations done by *R-Blur*.

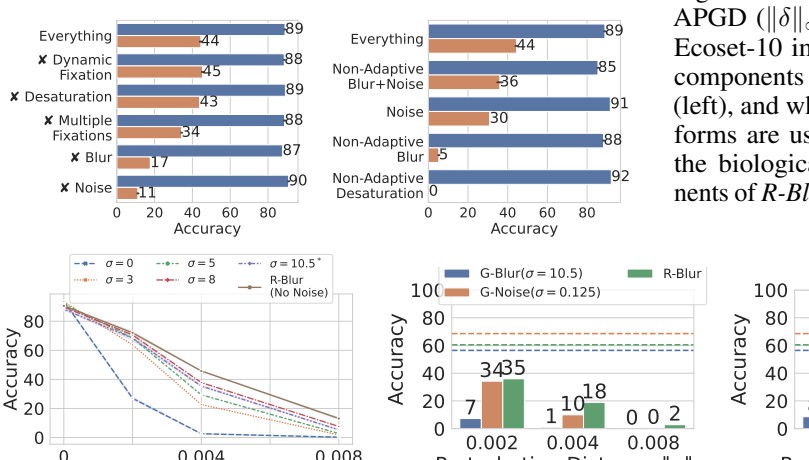

Figure 9: Accuracy on clean and APGD ($\|\delta\|_\infty = 0.008$) perturbed Ecoset-10 images when individual components of *R-Blur* are removed (left), and when non-adaptive transforms are used (right). Removing the biologically-motivated components of *R-Blur* harms the robustness

Figure 10: Comparison of *R-Blur* with models trained with non-adaptive Gaussian blur or Gaussian noise augmentations. (a) compares the accuracy under adversarial attack on Ecoset-10 of *R-Blur* and models augmented with non-adaptive Gaussian blur of various standard deviations ($\sigma$). While non-adaptive Gaussian blur does increase robustness, the adaptive blurring in *R-Blur* outperforms it by a margin. (b) compares the accuracy under adversarial attack on Imagenet of *R-Blur* and models augmented with either non-adaptive Gaussian blur or Gaussian noise. We see that *R-Blur* achieves better robustness than either of these methods.

### 3.3 Ablation Study

We examine, by way of ablation, how much each component of *R-Blur* contributes towards its overall robustness as shown in Figure 9. The most significant contributor to robustness is the addition of noise. This echoes the findings from [18], which showed that neural stochasticity contributes significantly to the robustness of the visual system. Nevertheless, even without noise *R-Blur* achieves an 11 point improvement over the vanilla ResNet which archives 0% accuracy under the attack, which indicates that other components of *R-Blur* also contribute towards robustness. Furthermore, experimental results reveal that robustness induced by noise diminishes as the complexity of the dataset increases and the size of the perturbations increases. As observed in Figure 10b, Gaussian noise augmentation achieves 45-58% (8-10 points) lower accuracy than *R-Blur*, and in Figure 7b, which shows that at larger perturbation sizes *R-Blur* achieves higher certified accuracy.

The second most significant contributor to robustness is the blurring performed by *R-Blur*. Importantly, we note that Gaussian blurring in and of itself does not greatly improve robustness. Figure 9 shows that non-adaptive blurring with a single Gaussian kernel having $\sigma = 10.5$ ($\sigma = 10.9$ is the maximum used in *R-Blur*) improves robustness by only 5 points. Furthermore, Figure 10a shows that increasing the strength of non-adaptive blurring trades off clean accuracy for robustness. However, after $\sigma = 8$ the gains in robustness hit (a rather low) ceiling, and increasing $\sigma$ further reduces both clean accuracy and robustness. On the other hand, *R-Blur*, *without any additive noise*, achieves similar clean accuracy as non-adaptive blurring ($\sigma = 10.5$) but achieves significantly better adversarial robustness, thereby demonstrating that Gaussian blurring alone accounts for only a fraction of *R-Blur*'s robustness. Furthermore, Figure 10b shows that the contribution of non-adaptive blurring declines on the more complex Imagenet, where it achieves only 1% accuracy on moderate-sized perturbations.

The next most significant factor, after noise and adaptive blurring, is evaluating multiple fixation points which improved robustness significantly compared to a single fixation point in the center of the image, which suggests that, multiple fixations and saccades are important when the image is hard to recognize and presents a promising direction for future work. Furthermore, not adaptively desaturating the colors reduces the robustness slightly. Finally, we note that dynamic fixation does not improve performance compared to 5 predefined fixation points. To summarize, most of the biologically-motivated components of *R-Blur* contribute towards improving the adversarial robustness of object recognition DNNs from close to 0% to 45% ($\ell_\infty = 0.008$ for Ecoset-10).

# 4 Related Work

**Non-biological defenses:** Perhaps the most successful class of adversarial defenses are adversarial training algorithms [7, 9, 49, 17, 8], which train models on adversarially perturbed data generated by backpropagating gradients from the loss to the input during each training step. Another popular class of defenses is certified defenses [10, 11, 50, 51] which are accompanied by provable guarantees of the form: with probability $1 - \delta$, the model's output will not change if a given image is perturbed at most $\epsilon$. Perhaps, most closely related to our work are preprocessing defenses [52, 53] that apply a large number of transforms to the input during inference. Usually, these defenses rely on non-differentiable transformations, and a high degree of randomization in the number, sequence, and parameters of the transforms they apply to each image. Therefore, these defenses tend to obfuscate gradients [47], and have been shown to be compromised by attacks with a higher step budget. We would like to point out that R-Blur does not have these aforementioned pitfalls – the transforms that R-Blur applies (Gaussian blur and desaturation) are fully differentiable and totally deterministic. In general, it is our opinion that by not being cognizant of the biological basis of robust vision, current approaches are excluding a large set of potentially effective approaches for defending against adversarial attacks.

**Biologically inspired defenses:** Several biological defenses have been proposed over the years. These defenses involve integrating computational analogues of biological processes that are absent from common DNNs, such as predictive/sparse coding [16, 17], biologically constrained visual filters, nonlinearities, and stochasticity [18], foveation [19, 20, 21, 22], into DNNs. The resulting models are made more robust to adversarially perturbed data, and have been shown to better approximate the responses of biological neurons [18].

Most relevant to our work are defenses that have integrated foveation with DNNs. One of the earliest works [20] implements foveation by cropping the salient region of the image at inference time. This work has several shortcomings. Firstly, the biological plausibility of this method is questionable because it does not simulate the degradation of visual acuity in the periphery of the visual field, rather it discards the periphery entirely. Secondly, it crops the image after applying the adversarial attack, which means that the attack does not take into account the cropping, which is akin to obfuscating the gradients, and hence any reported improvements in robustness are suspect. A later work [22] (*R-Warp*) avoids the aforementioned pitfalls and simulates foveation via non-uniform sampling (regions further away from the fixation points are sampled less densely). Since this method is fully differentiable and highly biologically plausible, we compare against it in this paper. Some recent works [19, 21] apply foveation in the latent feature space (the intermediate feature maps generated by a CNN). These works implement foveation by changing the receptive field sizes of the convolutional kernels based on the distance to the fixation. Since they operate on the latent feature space, rather than image pixels, their methods not directly comparable to ours.

# 5 Limitations

Adding *R-Blur* reduces accuracy on clean data, however, it is possible to significantly improve the accuracy of *R-Blur* by developing better methods for selecting the fixation point. Further experimental results presented in Appendix B show that if the optimal fixation point was chosen by an oracle the clean accuracy of *R-Blur* can be improved to within 2% of the accuracy of the unmodified ResNet.

# 6 Conclusion

Since the existence of adversarial attacks presents a divergence between DNNs and humans, we ask if some aspect of human vision is fundamental to its robustness that is not modeled by DNNs. To this end, we propose *R-Blur*, a foveation technique that blurs the input image and reduces its color saturation adaptively based on the distance from a given fixation point. We evaluate *R-Blur* and other baseline models against APGD attacks on two datasets containing real-world images. *R-Blur* outperforms other biologically inspired defenses. Furthermore, *R-Blur* also significantly improves robustness to common, non-adversarial corruptions and achieves accuracy greater than that of adversarial training. The robustness achieved by *R-Blur* is certifiable using the approach from [10] and the certified accuracy achieved by *R-Blur* is at par or better than that achieved by randomized smoothing [10]. Our work provides further evidence that biologically inspired techniques can improve the accuracy and robustness of AI models.

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

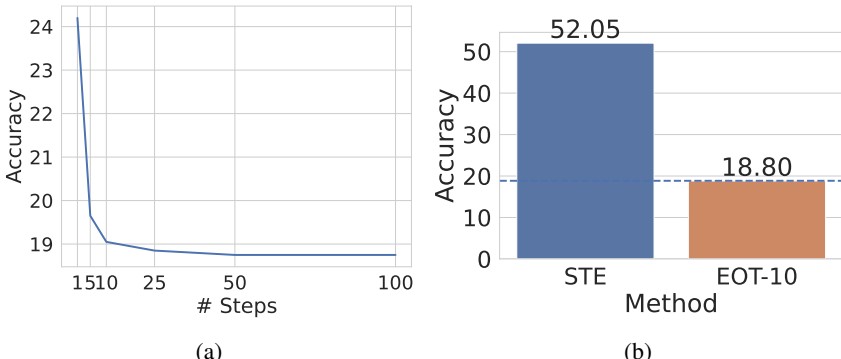

(a)                                          (b)

Figure 11: Accuracy of a *R-Blur* model trained on Imagenet under APGD attack with different settings. (a) shows the accuracy when APGD attack is applied with different numbers of update steps. (b) shows the accuracy when 10 step of expectation-over-transformation (EOT-10) [54] is used and *R-Blur* is converted into a straight-through-estimator (STE) in the backward pass. The dashed line in (b) shows the accuracy of a 25-step APGD attack without EOT and normal gradient computation for *R-Blur*. Together these results strongly indicate that *R-Blur* does not obfuscate gradients and legitimately improves the adversarial robustness of the model.

## Appendix

## A   Preventing Gradient Obfuscation

We take a number of measures to ensure that our results correspond to the true robustness of our method, and we avoid the pitfalls of gradient obfuscation [47, 48].

Firstly, we remove inference time stochasticity from all the models we test. We do this by sampling the Gaussian noise used in *R-Blur* and *VOneBlock* once and applying the same noise to all test images. Similarly, we sample the affine transform parameters for *RandAffine* once and use them for all test images. We also compute the fixation point sequences for *R-Blur* and *R-Warp* on unattacked images and do not update them during or after running APGD.

Secondly, we ran APGD for 1 to 100 iterations and observed that as the number of iterations increases the success rate of the attack increases (Figure 11a). The success rate plateaus at 50 iterations. Since the attack success rate with 25 steps is only 0.1% lower than the success rate with 50 steps, we run APGD with 25 steps in most of our experiments.

Thirdly, we evaluate *R-Blur* against AutoAttack [46], an ensemble of 4 state-of-the-art white and black box adversarial attacks. Figure 12 compares the accuracy of *R-Blur* on Imagenet under APGD and AutoAttack. We see that the accuracy under AutoAttack is only slightly lower than the accuracy under APGD, with the maximum difference being 3%, which would not change any of the trends observed in the paper. Since computing AutoAttack requires a lot of time and compute, and given that it does not decrease accuracy significantly compared to 25-step APGD, we chose to use the latter for most of the results presented in the paper.

Finally, we applied expectation over transformation [54] by computing 10 gradient samples at each APGD iteration and averaging them to obtain the final update. We found this did not change the attack success rate so we take only 1 gradient sample in most of our experiments (Figure 11b). Finally, we also used a straight-though-estimator to pass gradients through *R-Blur* in case it may be obfuscating them and found that doing so reduces the attack success rate, thus indicating that gradients that pass through *R-Blur* retain valuable information that can be used by the adversarial attack (Figure 11b).

## B   Fixation Point Selection

In this study, we did not attempt to develop an optimal fixation point selection algorithm, and instead, we operate under the assumption that points at which humans tend to fixate are sufficiently informative to perform accurate object classification. Therefore, we used DeepGaze-III [44], which is a neural

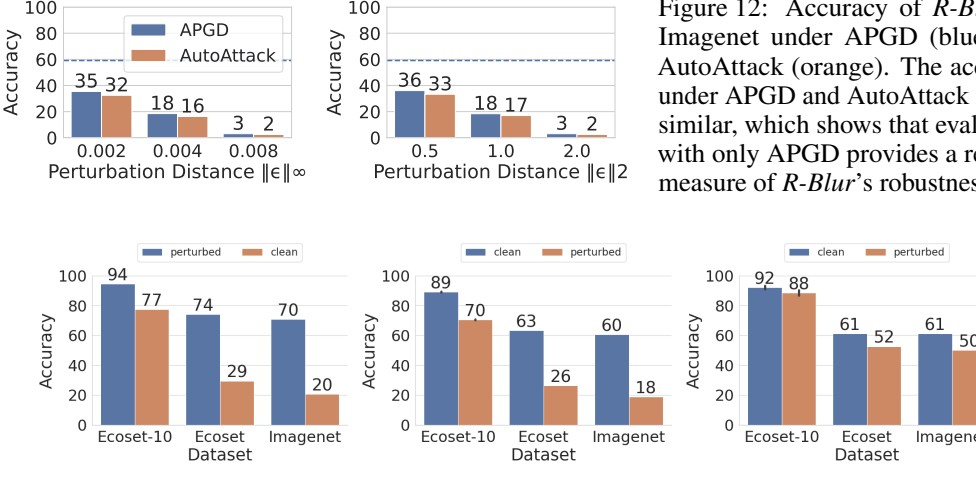

Figure 12: Accuracy of *R-Blur* on Imagenet under APGD (blue) and AutoAttack (orange). The accuracy under APGD and AutoAttack is very similar, which shows that evaluating with only APGD provides a reliable measure of *R-Blur*'s robustness

(a) Optimal Fixation   (b) DeepGaze-III Five Fixations   (c) Adversarial Training

Figure 13: The accuracy obtained on clean and adversarial data when (a) the optimal fixation point was selected, (b) when the five fixation approach from Section 3 was used, and (c) an adversarially trained model was used.

network model trained to model the human gaze. DeepGaze-III uses a deep CNN backbone to extract features from the image, and based on these features another DNN predicts a heatmap that indicates, for each spatial coordinate, the probability that a human will fixate on it. However, it is possible that this algorithm is sub-optimal, and with further study, a better one could be developed. Though developing such an algorithm is out of the scope of this paper, we conduct a preliminary study to determine if it is possible to select better fixation points than the ones predicted by DeepGaze-III.

To this end, we run the following experiment to pick an optimal fixation point for each image during inference. For each testing image, we select 49 fixation points, spaced uniformly in a grid. Using the models we trained in earlier (see section 3) we obtain predictions for each image and each of the 49 fixation points. If there was at least one fixation point at which the model was able to correctly classify the image, we consider it to be correctly classified for the purpose of computing accuracy. We repeat this experiment for Ecoset-10, Ecoset, and Imagenet, using clean and adversarially perturbed data. We obtain the adversarially perturbed images for each of the 49 fixation points by fixing the fixation point at one location running the APGD attack with $\ell_\infty$-norm bounded to 0.004. Figure 14 illustrates this experiment with some example images.

The results are presented in Figure 13. We see that when the optimal fixation point is chosen accuracy on both clean and adversarially perturbed data improves, with the improvement in clean accuracy being the most marked. The clean accuracy on Ecoset-10, Ecoset, and Imagenet improved by 5%, 11%, and 10% respectively, which makes the clean accuracy of the *R-Blur* model on par or better than the clean accuracy achieved by the unmodified ResNet. Furthermore, when the optimal fixation point, is chosen *R-Blur* obtains higher clean accuracy than *AT* on all the datasets.

These results are meant to lay the groundwork for future work toward developing methods for determining the optimal fixation point based on the input image. However, they also illustrate that models trained with *R-Blur* learn features that are not only more adversarially robust features than ResNet but also allow the model to make highly accurate predictions on clean data.

## C   Evaluations With Different Architectures

To demonstrate that the benefits of *R-Blur* are not limited to CNNs, we trained MLP-Mixer [55] and ViT [56] models with *R-Blur* preprocessing and evaluated their robustness. We use the configuration of MLP-Mixer referred to as S16 in [55]. Our ViT has a similar configuration, with 8 layers each having a hidden size of 512, an intermediate size of 2048, and 8 self-attention heads. We train both models with a batch size of 128 for 60 epochs on Ecoset-10 using the Adam optimizer. The learning

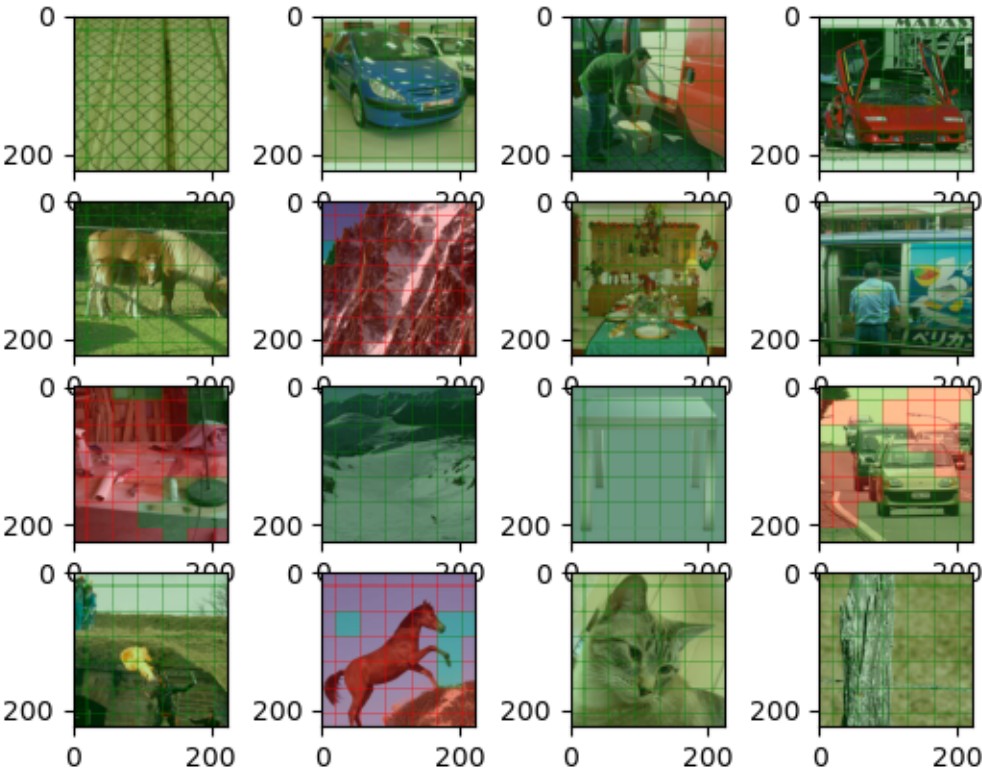

Figure 14: This figure indicates the locations of the optimal fixation points for some sample images. Each square in the grid corresponds to one of 49 fixation locations and represents the highest resolution region of the image if the model fixates at the center of the square. Squares that are shaded green indicate that the model's prediction at the corresponding fixation point was correct, while squares shaded red indicate that the model's prediction at the corresponding fixation point was incorrect. We see that there are certain images in which there are only a few optimal fixation points and they may not be in the center or in the corners of the image.

rate of the optimizer is linearly increased to 0.001 over 12 epochs and is decayed linearly to almost zero over the remaining epochs. The results are shown in Figure 15.

We observe that *R-Blur* significantly improves the robustness of MLP-Mixer models, and achieves greater accuracy than *R-Warp* at higher levels of perturbations. These results show that the robustness endowed to ResNets by *R-Blur* was not dependent on the model architecture, and they further strengthen our claim that loss in fidelity due to foveation contributes to the robustness of human and computer vision.

## D    Breakdown of Accuracy Against Common Corruption by Corruption Type

In Figure 16 we break down the performance of the models on common corruptions by higher-level corruption categories. The individual members of each category are listed in Table 2. We see that in most of the categories, *R-Blur* achieves the highest median accuracy against the most severe corruptions. We also note that *R-Blur* exhibits a remarkable degree of robustness to noise, which is substantially greater than all the other models we evaluated. It is pertinent to note here that Gaussian noise was just 1 of the 4 types of noise included in the noise category, and thus the performance of *R-Blur* can not be attributed to overfitting on Gaussian noise during training. Furthermore, robustness to

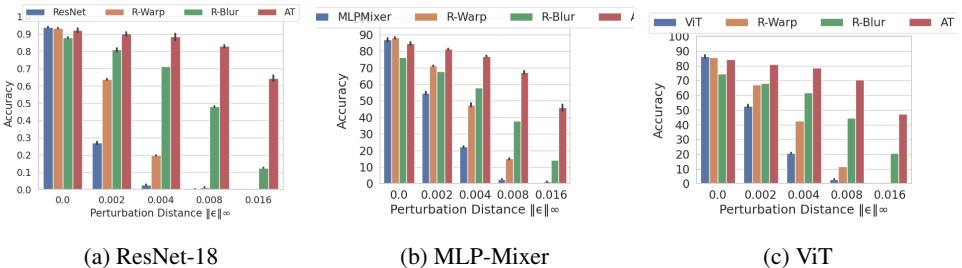

|             | (a) ResNet-18 | (b) MLP-Mixer | (c) ViT |
|-------------|---------------|---------------|---------|

Figure 15: The accuracy obtained on Ecoset-10 against adversarial perturbations of various $\ell_\infty$ norms when *R-Blur* is used with ResNet, MLP-Mixer and ViT backbones.

| Noise | Blur | Weather | Digital |
|-------|------|---------|---------|
| gaussian noise | defocus blur | snow | contrast |
| shot noise | glass blur | frost | elastic transform |
| impulse noise | motion blur | fog | pixelate |
| speckle noise | zoom blur | brightness | jpeg compression |
|  | gaussian blur | spatter | saturate |

Table 2: Categories of corruptions used to evaluate robustness to common corruptions. This categorization follows the one from [38]

one type of random noise does not typically generalize to other types of random noise [4]. Therefore, the fact that *R-Blur* exhibits improved robustness to multiple types of noise indicates that it is not just training on Gaussian noise, but rather the synergy of all the components of *R-Blur* that is likely the source of its superior robustness.

# E    Sensitivity Analysis of Hyperparameters in *R-Blur*

To measure the influence of the various Hyperparameters of *R-Blur* we conduct a sensitivity analysis. First, we vary the scale of the Gaussian noise added to the image, the viewing distance during inference, and the value of $\beta$ from Section 2.5, which is the scaling factor that maps eccentricity (see equation 1 to standard deviation, and measure the impact on accuracy on clean as well as adversarially perturbed data. The results of this analysis are presented in Figure 17. We see that, as expected, increasing the scale of the noise improves accuracy on adversarially perturbed data, however, this improvement does not significantly degrade clean accuracy. It appears that the adaptive blurring is mitigating the deleterious impact of Gaussian noise on clean accuracy. On the other hand, increasing $\beta$ beyond 0.01 surprisingly does not have a significant impact on accuracy and robustness. We also measured the accuracy on clean and perturbed data after varying the viewing distance (see 2.4) and the number of fixation points over which the logits are aggregated. These results are plotted in Figure 18, and they show that accuracy on clean and perturbed data is maximized when the width of the in-focus region is 48 (this corresponds to $vd = 3$) and aggregating over more fixation points improves accuracy on clean and perturbed data.

# F    Training Configuration

Table 3 presents the configurations used to train the models used in our evaluation. For all the models the SGD optimizer was used with Nesterov momentum=0.9.

# G    Implementation Details

We used Pytorch v1.11 and Python 3.9.12 to for our implementation. We used the implementation of Auto-PGD from the Torchattacks library (https://github.com/Harry24k/adversarial-attacks-pytorch). For *R-Warp* we used the code from the official repo https://github.com/mvuyyuru/adversary.git. Likewise, for *VOneBlock* we used the code from https://github.com/dicarlolab/vonenet, and

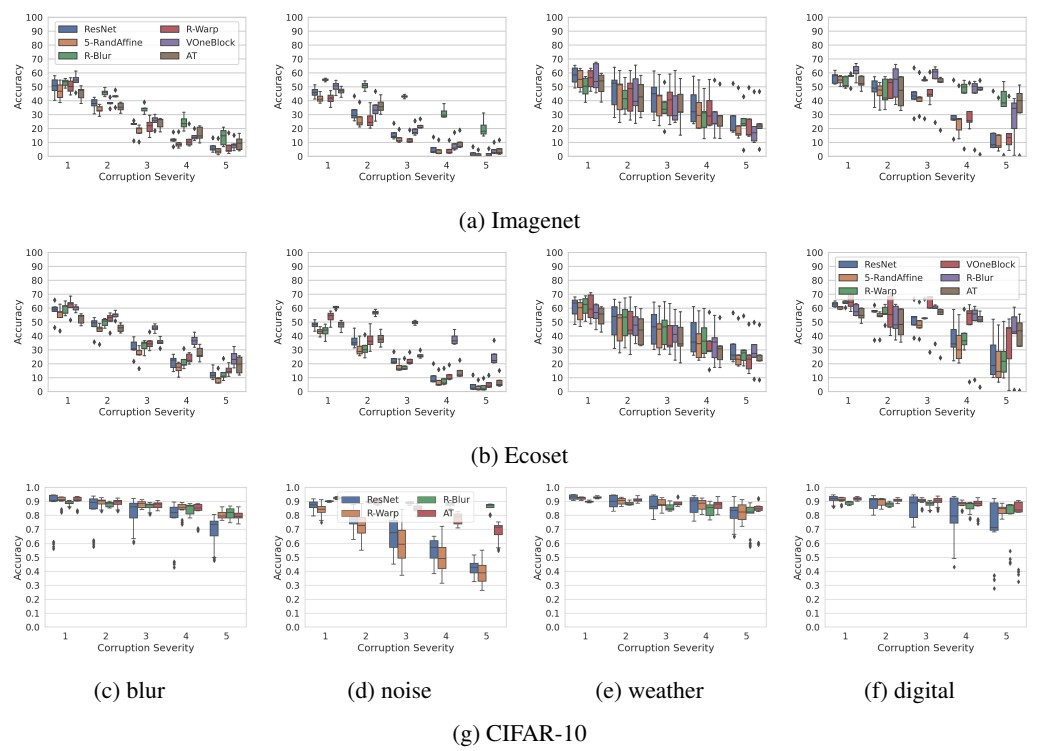

(a) Imagenet

(b) Ecoset

(c) blur       (d) noise       (e) weather       (f) digital

(g) CIFAR-10

Figure 16: The accuracy achieved by *R-Blur* and baselines on various classes of common corruptions, proposed in [38]. The boxplot shows the distribution of accuracy values on 4-5 different corruptions in each class applied at different severity levels (x-axis) with 1 referring to least severe and 5 being the most severe corruption. *R-Blur* generally achieves the highest median accuracy on the highest severity levels.

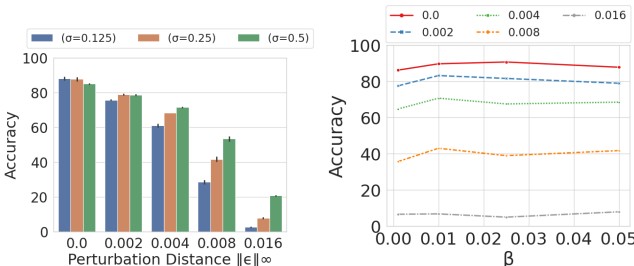

Figure 17: The impact of the hyperparameters of *R-Blur* on the accuracy and robustness of models trained on Ecoset-10. (left) the standard deviation of Gaussian noise, and (right) $\beta$ from Section 2.5.

for DeepGaze-III models we used the code from https://github.com/matthias-k/DeepGaze. The training code for DeepGaze-III with *R-Blur* and *R-Warp* backbones is based on https://github.com/matthias-k/DeepGaze/blob/main/train_deepgaze3.ipynb, and can be found in `adversarialML/biologically_inspired_models/src/fixation_prediction/train_deepgaze.py`. Our clones of these repositories are included in the supplementary material. For multi-gpu training, we used Pytorch Lightning v1.7.6. We used 16-bit mixed precision training to train most of our models. The code for *R-Blur* can be found in `adversarialML/biologically_inspired_models/src/retina_preproc.py` which is part of the supplemental material.

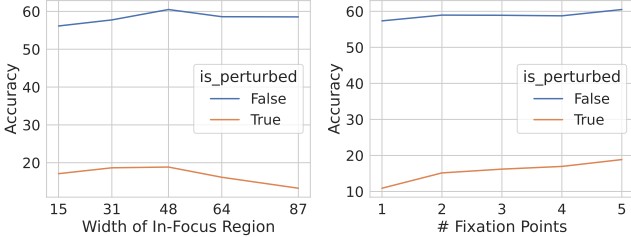

Figure 18: The impact of the size of the in-focus region by varying the viewing distance (left) and the number of fixation points over which the logits are aggregated (right) on accuracy. The plots are computed from a *R-Blur* model trained on Imagenet, and the perturbed data is obtained by conducting a 25-step APGD attack with $\|\delta\|_\infty = 0.004$. We see that accuracy on clean and perturbed data is maximized when the width of the in-focus region is 48 (this corresponds to $vd = 3$) and aggregating over more fixation points improves accuracy on clean and perturbed data.

| Dataset | Method | Batch Size | nEpochs | LR | LR-Schedule | Weight Decay | nGPUs |
|---------|--------|-----------|---------|-----|-------------|--------------|-------|
| CIFAR-10 | ResNet | 128 | 0.4 | 60 | L-Warmup-Decay(0.2) | 5e-5 | 1 |
| | AT | 128 | 0.4 | 60 | L-Warmup-Decay(0.2) | 5e-5 | 1 |
| | R-Warp | 128 | 0.4 | 60 | L-Warmup-Decay(0.2) | 5e-5 | 1 |
| | R-Blur | 128 | 0.4 | 60 | L-Warmup-Decay(0.2) | 5e-5 | 1 |
| | G-Noise | 128 | 0.4 | 60 | L-Warmup-Decay(0.2) | 5e-5 | 1 |
| Ecoset-10 | ResNet | 128 | 0.4 | 60 | L-Warmup-Decay(0.2) | 5e-4 | 1 |
| | AT | 128 | 0.4 | 60 | L-Warmup-Decay(0.2) | 5e-4 | 1 |
| | R-Warp | 128 | 0.4 | 60 | L-Warmup-Decay(0.2) | 5e-4 | 1 |
| | R-Blur | 128 | 0.1 | 60 | L-Warmup-Decay(0.1) | 5e-4 | 1 |
| | VOneBlock | 128 | 0.1 | 60 | L-Warmup-Decay(0.1) | 5e-4 | 1 |
| | G-Noise | 128 | 0.4 | 60 | L-Warmup-Decay(0.2) | 5e-4 | 1 |
| Ecoset | ResNet | 256 | 0.2 | 25 | L-Warmup-Decay(0.2) | 5e-4 | 2 |
| | AT | 256 | 0.2 | 25 | L-Warmup-Decay(0.2) | 5e-4 | 4 |
| | R-Warp | 256 | 0.1 | 25 | L-Warmup-Decay(0.2) | 5e-4 | 4 |
| | R-Blur | 256 | 0.1 | 25 | C-Warmup-2xDecay(0.1) | 5e-4 | 4 |
| | VOneBlock | 256 | 0.1 | 25 | C-Warmup-2xDecay(0.1) | 5e-4 | 4 |
| | G-Noise | 256 | 0.1 | 25 | C-Warmup-2xDecay(0.1) | 5e-4 | 4 |
| Imagenet | ResNet | 256 | 0.2 | 25 | L-Warmup-Decay(0.2) | 5e-4 | 2 |
| | AT | 256 | 0.2 | 25 | L-Warmup-Decay(0.2) | 5e-4 | 4 |
| | R-Warp | 256 | 0.1 | 25 | L-Warmup-Decay(0.2) | 5e-4 | 4 |
| | R-Blur | 256 | 0.1 | 25 | C-Warmup-2xDecay(0.1) | 5e-4 | 4 |
| | VOneBlock | 256 | 0.1 | 25 | C-Warmup-2xDecay(0.1) | 5e-4 | 4 |
| | G-Noise | 256 | 0.1 | 25 | C-Warmup-2xDecay(0.1) | 5e-4 | 4 |

Table 3: The configurations used to train the models used in our evaluation. L-Warmup-Decay($f$) represents a schedule that linearly warms up and decays the learning rate and $f$ represents the fraction of iterations devoted to warmup. C-Warmup-2xDecay(0.1) is similar except that the warmup and decay follow a cosine function, and there are two decay phases. Both the schedulers are implemented using `torch.optim.lr_scheduler.OneCycleLR` from Pytorch.

| Method | Dynamic Fixation | # Fixations | Train Speed (img/s) | Test Speed (img/s) |
|:---:|:---:|:---:|:---:|:---:|
| ResNet | ✗ | 1 | 410 | 370 |
| *AT* | ✗ | 1 | 232 (1.8×) | - |
| *VOneBlock* | ✗ | 1 | 277 (1.5×) | 289 (1.3×) |
| *R-Warp* | ✗ | 1 | 377 (1.1×) | 314 (1.2×) |
| *R-Blur* | ✗ | 1 | 369 (1.1×) | 334 (1.1×) |
| *R-Blur* | ✗ | 5 | - | 111 (3.3×) |
| *R-Warp* | ✓ | 5 | - | 26 (14.2×) |
| *R-Blur* | ✓ | 1 | - | 115 (3.2×) |
| *R-Blur* | ✓ | 5 | - | 29 (12.9×) |

Table 4: Training and inference speed on 1 Nvidia 2080Ti, measure in images per second, for each model. The value in the parentheses indicates the slowdown relative to the unmodified ResNet computed as the (train/test) speed of the ResNet divided by the speed of the other method. We see that *R-Blur*, in and of itself, causes a very minimal slowdown (only $1.1\times$) during training and testing. Increasing the number of fixations slows R-Blur only sub-linearly (5 predefined fixations lead to 3x slowdown). Introducing dynamic fixation prediction has a greater impact on speed because each image is assigned different fixation points and so R-Blur/R-Warp can not be applied to them as a single batch. This shortcoming is likely common to most fixation transforms, and is not unique to R-Blur.

## H  Hardware Details and Computation Cost

We trained our models on compute clusters with Nvidia GeForce 2080 Ti and V100 GPUs. Most of the Imagenet and Ecoset models were trained and evaluated on the V100s, while the CIFAR-10 and Ecoset-10 models were trained and evaluated on the 2080 Ti's.

### H.1  Analysis of Computation Cost

Table 4 presents this comparison and shows that R-Blur causes minimal slowdown (1.1x compared to the vanilla ResNet) during both training and testing. Also, increasing the number of fixations slows R-Blur only sub-linearly (5 predefined fixations => 3x slowdown). Introducing dynamic fixation prediction has a greater impact on speed because each image is assigned different fixation points and so R-Blur/R-Warp can not be applied to them as a single batch. This shortcoming is likely common to most fixation transforms, and is not unique to R-Blur. In fact, under dynamic fixation prediction, R-Blur is faster than R-Warp.

