# OpenReview forum: "Training on Foveated Images Improves Robustness to Adversarial Attacks"
_NeurIPS.cc/2023/Conference — NeurIPS 2023 poster_

### Official Review · Reviewer_atHR · 2023-06-30

**Soundness:** 3 good
**Presentation:** 4 excellent
**Contribution:** 3 good
**Rating:** 7
**Confidence:** 5

**Summary:**

This paper studies the effect of foveation via adaptive gaussian blurring and color modulation in training on an image. Their goal is not computer vision driven, nor ML based, but rather to shed light on the physiological nature of the retina and spatially adaptive computation in humans -- that may prove useful for machines towards achieving robustness without adversarial training.

**Strengths:**

* This paper was very easy to read, and understand. All the figures and tables are coherent, so I applaud the efforts of presentation of the authors.
* Authors co-modulate adaptive gaussian blurring with color loss and noise (however the contribution of each factor in the "foveation" is not obvious).
* Authors present a solid list of experiments in o.o.d. experiments and also adversarial attacks. Their benchmarks regarding adversarial training are on spot (though which type of AT I believe is not precised).
* Their claims are grounded, and not too strong, so that is good given the evidence they have presented and also works that other authors have done.

**Weaknesses:**

I believe the strongest weakness is that it's not clear to me what the contribution is of each effect : 1) noise ; 2) adaptive gaussian blurring; 3) fixations.

I wonder to what point do the extra fixations add as a proxy for data augmentation when training the model. I.e.: can a false conclusion be arrived that "foveation aids in robustness" by virtue of adding extra images. I think authors need to add 2 more controls for me to be more enthusiastic about this paper:
1) Train all non-foveated networks with additional sets of images via data augmentation procedures (like rotation, mirroring, random crop and blurring)
2) Train all foveated networks with less images to match the original dataset size of the networks that recieve non-foveated inputs.

In addition it's not clear to me why authors add noise as part of the foveation. Why not just go straight towards the adaptive gaussian blurring? Is there a psychological reason to add the noise in the foveation process?

**Questions:**

* Missing References (to add in line 32 and through-out the paper regarding links of biological vision (peripheral computation) + adversarial robustness):
**“Finding Biological Plausibility for Adversarially Robust Features via Metameric Tasks”. Harrington & Deza. ICLR 2022.**

* Missing Reference in Line 51: I believe Pramod et al. did in fact perform robustness tests on their foveated adaptive blur image.

* Missing Reference in Line 52: Characterizing a snapshot of perceptual experience. By Cohen et al. from the Journal of Experimental Pscyhology 2021.

Overall I think this paper is a good contribution to the field, it should provide further discussion at NeurIPS about foveation, which many vision scientists have been longing to see these type of augmentations be used in computer vision. It would be quite nice if authors release a R-Blur augmentatino module in pytorch.

**Limitations:**

See Weaknesses above.

---

> ### Author Rebuttal · Authors · 2023-08-09
>
> We thank the reviewer for providing valuable feedback on our paper, and asking thoughtful questions. We are glad and encouraged to find that the reviewer liked our work and found it to be unique and interesting. We hope that in our responses below we will be able to fully address the reviewer's outstanding concerns and further improve the reviewer's opinion of our work.
>
> **I believe the strongest weakness is that it's not clear to me what the contribution is of each effect : 1) noise ; 2) adaptive gaussian blurring; 3) fixations.**
>
> We agree with the reviewer that it is essential to gauge the impact of the different components of R-Blur, and therefore we have conducted an ablation study and presented it in Section 3.3 and Figure 8. To ascertain the impact of each component of R-Blur we removed or disabled each component one by one, while leaving all the other components in place. We find that removing the noise led to the greatest reduction in accuracy under adversarial attack (-33%), followed by the removal of adaptive blurring (-27%) and using 1 fixation instead of 5 (-10%). We hope that this addresses the reviewer’s concerns. We also encourage the reviewer to refer to section 3.3 for more details. If our analysis is lacking in any way we would be eager to further discuss this with the reviewer and do everything possible to improve it.
>
> **[...] can a false conclusion be arrived that "foveation aids in robustness" by virtue of adding extra images. I think authors need to add 2 more controls for me to be more enthusiastic about this paper: (1)Train all non-foveated networks with additional sets of images via data augmentation procedures (like rotation, mirroring, random crop and blurring. (2)  Train all foveated networks with less images to match the original dataset size of the networks that recieve non-foveated inputs.**
>
> These tests are, in fact, reported, although, perhaps we did not communicate this clearly in our writing. Specifically, we train all our models (those with R-Blur and those without) with RandAugment or AutoAugment, which includes the data augmentations mentioned by the reviewer in addition to a wide variety of other non-geometric transformations. While we have indicated this in section 3.1 (Ln165-166), we will try to make it more explicit that the data augmentations are the same for all the models.
>
> Under these conditions we believe that the current evaluation set is fair. Apart from the baselines, all the models are trained with the same number of data augmentations, that is RandAugment + (PGD attack/R-Warp/VOneBlock/R-Blur). Nevertheless, we would be happy to further discuss with the reviewer if there is something that we missed and would try to make the evaluation as fair as possible.
>
> **it's not clear to me why authors add noise as part of the foveation.**
>
> We add noise to simulate the stochasticity in the responses of the biological neurons in the retina (Croner, et.al 1993). We seem to have omitted this citation from the paper, but we will update the paper to include it.
>
> Croner LJ, Purpura K, Kaplan E. Response variability in retinal ganglion cells of primates. Proc Natl Acad Sci U S A. 1993 Sep 1;90(17):8128-30. doi: 10.1073/pnas.90.17.8128. PMID: 8367474; PMCID: PMC47301.
>
> **Missing References (to add in line 32 and through-out the paper regarding links of biological vision (peripheral computation) + adversarial robustness): “Finding Biological Plausibility for Adversarially Robust Features via Metameric Tasks”. Harrington & Deza. ICLR 2022.**
>
> We thank the reviewer for pointing us to this paper, we will cite this in the paper.
>
> **Missing Reference in Line 51: I believe Pramod et al. did in fact perform robustness tests on their foveated adaptive blur image.**
>
> We were unable to find robustness experiments in [29]. We referred to the version on Arxiv which is also the one we cited, but we also looked at the version on PubMed. If we overlooked or misunderstood something in [29] we would be happy to be corrected.
>
>
> **Missing Reference in Line 52: Characterizing a snapshot of perceptual experience. By Cohen et al. from the Journal of Experimental Pscyhology 2021.**
>
> We thank the reviewer for pointing us to this paper, we will cite this in the paper.
>
> **It would be quite nice if authors release a R-Blur augmentatino module in pytorch.**
> PyTorch code is included in the supplementary material and will be made public on GitHub after publication.
>
> We hope that our responses fully address the reviewer's concerns and we kindly request the reviewer to consider increasing their score of our paper.

---

> > ### Comment · Reviewer_atHR · 2023-08-17
> > **Thank you for addressing my comments | Worth high-lighting role of Noise**
> >
> > Thank you for addressing my concerns. I will increase my score from 6 to 7.
> >
> > I think what definitely needs to be high-lighted (as another review er pointed out) is the role of noise in this framework. There is a paper I believe also cited in the submission by Dapello, Marques et al. NeurIPS 2020 (Simulating a Primary Visual Cortex at the Front of CNNs Improves Robustness to Image Perturbations) that has a similar story where most of the job is being done by the noise rather than the Gabors, so all-in-all authors should find a way to add stress this in the discussion so that future works should also analyze the co-modulation of noise with certain inductive biases and how they affect adversarial robustness in DNNs (to see what property is carrying the robustness weight).

---

> > > ### Author Response · Authors · 2023-08-19
> > >
> > > We are glad that we were able to fully address the reviewer's concern, and we thank them for increasing their score. The reviewer's recommendation is well taken and we will definitely highlight the role of noise in the camera-ready version.

---

### Official Review · Reviewer_ysro · 2023-07-05

**Soundness:** 3 good
**Presentation:** 3 good
**Contribution:** 3 good
**Rating:** 7
**Confidence:** 3

**Summary:**

In their paper the authors introduce R-Blur as a foveation technique for biologically inspired defense against adversarial attacks on DNNs.
With their approach they try to simulate the human visual field by blurring and desaturating the image depending on the distance to a given fixation point. They continue to evaluate their approach on Ecoset and ImageNet and try to show that R-Blur can achieve results like adversarial training. Furthermore, they provide an ablation study to prove the significance of the individual components.

**Strengths:**

- I find the idea of taking a biologically inspired approach to foveation and the attempt to recreate the human perceptive field through color and grey acuity appealing.
- The results for the common non-adversarial corruption show an improvement compared to other methods indicating some significance.


**Weaknesses:**

- In Figure 5 we only see a comparison with two baselines that feature little to no adversarial defense. Here it is clear that R-Blur achieves higher accuracies under white-box attacks. A comparison with adversarial training would be more insightful, because later only the mean value of these experiments is presented which indicates that AT outperforms R-Blur by a margin. This weakens the statement that R-Blur generalizes robustness.

- The ablation study provides some insights into the importance of some components but is not explained adequately, e.g., the dynamic selection of the fixation point seems to worsen the performance.

- The formatting to indicate the best and the second-best entries of Table 1 is somewhat misleading: it would be better to underline the second-best result or use colors.

- Writing: Some of the figure captions are not comprehensible or not descriptive enough (e.g., Figure 2 and Figure 3).

Overall: Although R-Blur is well-motivated and implemented in a reasonable way, only a small part of the experiments indicates a significant improvement (mainly the non-adversarial corruptions, Table 1.) while in general adversarial training seems to be superior. Currently I decide for a borderline reject, but if the before mentioned weaknesses are addressed and the following questions are answered, I am willing to increase the rating.

Minor Detail: Some figures (e.g., Figure 7 and especially Figure 2) are too small and are sometimes missing entries (e.g., R-Blur-5FL in Figure 7(b).1 or 0 entries in Figure 5)

**Questions:**

- The main motivation for the parameter setting for the visual acuity estimation is based on the photopic and visual acuity in human vision which should approximate the real curves (Figure 1(a)). It is not entirely clear how this estimated acuity relates to the original, because there it is measured on the x-axis through the degree in the visual field. In Figure 1 it seems to be the distance-based eccentricity although it should be zero were the original had zero degrees. How exactly does your approximation match the original and why does the transfer work?

- In the related work section several other foveation-based robustness models are mentioned and only a very short comparison is given. A more detailed comparison with other foveation techniques in general would be insightful as well. It is mentioned that these techniques were not used in an adversarial context before, but how does R-Blur stand out?

- For the eccentricity computation only on maximum of the Manhattan-Norm was considered. Were other quantified metrics (e.g., L2-Norm as in the actual human perceptive field) used in the experiments and how did they perform?

**Limitations:**

The authors address their limitations adequately in their Limitations section. These include a significant loss of clean accuracy and the current fixpoint selection method.

---

> ### Author Rebuttal · Authors · 2023-08-10
>
> We thank the reviewer for providing valuable feedback on our paper. We find it encouraging that the reviewer will earnestly consider improving their rating if their questions are answered.
>
> **In Figure 5 [...] A comparison with adversarial training would be more insightful**
>
> The goal of Figure 5 is to show that simulating foveation with R-Blur improves robustness. This can be broken into two sub-goals: (1) showing that augmenting a model with R-Blur improves its adversarial robustness (ResNet vs R-Blur), and (2) showing that the improvement in robustness is due to simulating foveation and not due to simply applying any arbitrary transforms at inference time (RandAffine vs. R-Blur).  Furthermore, we do not claim that R-Blur is a SOTA adversarial defense, and, therefore, we do not position it as a competitor of AT as far as whitebox adversarial attacks are concerned. Therefore, including AT in Figure 5 would only convolute the message it was meant to convey.
>
> However, at the reviewer’s request, we have created Figure 1 in the global response that compares the accuracy of R-Blur and AT at different adversarial perturbation sizes. Likewise, figure 5 in the appendix shows the breakdown of the accuracy of all the models against common corruptions of various types and strengths.
>
> **[...]AT outperforms R-Blur by a margin. This weakens the statement that R-Blur generalizes robustness.**
>
> We would like to point out that AT outperforms R-Blur only on whitebox attacks but not on non-adversarial corruptions. It is expected that AT models will be more robust to whitebox attacks. This is because they are trained on adversarial attacks that are very similar to the attacks used during testing, and therefore  the gap between the training and test distribution of AT models is much smaller than models like R-Blur which were not trained on adversarially perturbed data.
>
> For this reason, we state in Ln255-256 that we are comparing AT and R-Blur primarily in terms of their robustness to non-adversarial image perturbations. From Table 1 we see that AT has almost no impact on robustness to non-adversarial image perturbations. On the other hand, R-Blur improves the robustness of the model, not only to adversarial attacks but also to non-adversarial perturbations. These results show that the robustness of AT is limited to a small class of perturbations generated by norm-bounded adversarial attacks, while the robustness of R-Blur generalizes better across different types of perturbations.
>
> We acknowledge that a lack of clarity in our presentation might have given rise to this misunderstanding. Perhaps the mean scores in columns 2-4 are not necessary and are obscuring the point we intend to convey. We will consider removing them.
>
> ** importance of dynamic fixation selection not explained adequately in ablation**
>
> We had decided to not discuss the dynamic fixation selection in the text for two reasons: (1) the fixation selection model (DeepGaze-III) is not part of R-Blur and we only use it as a tool, and (2) Figure 8 indicates that switching between dynamic and static fixations trades off <= 1% accuracy for robustness, however verifying this trade-off would require more experimentation which we felt was out of the scope of this paper. Nevertheless, we will briefly discuss these points in the text.
>
> **[...]In Figure 1 it seems to be the distance-based eccentricity[...]**
> We request the reviewer to please correct us if we’re wrong but it seems that the reviewer is referring to Figure 2, not Figure 1. The x-axis in Figure 2 is indeed mislabeled and is showing the index of the pixel on a horizontal line through the fixation point. We have updated this figure to reflect the eccentricity as computed by equation 1. We will include this figure in the paper and have also included it in the global response (Figure 2).
>
> **[...]A more detailed comparison with other foveation techniques[...].**
>
> We will include the following comparison in the paper.
>
> [19,20,21,22] claim to simulate some aspect of foveation. [19] implemented foveation by cropping the salient region of the image at inference time. Firstly, the biological plausibility of [19] is questionable because instead of simulating the degradation of acuity in the periphery of the visual field, it simply discards it. Secondly, they crop the image _after_ applying the adversarial attack, which likely obfuscates the gradients, and hence any robustness they report is suspect. On the other hand, [20] and [21] apply foveation in the latent feature space (the intermediate feature maps generated by a CNN) rather than directly to the image pixels as we do so their methods are not directly comparable to ours. To the best of our knowledge [22] (R-Warp) is the only foveation-based adversarial defense that is biologically plausible, works directly on the pixels, and avoids gradient obfuscation, which is why we compare against it in this paper.
>
> Furthermore, in Ln50-52 we do acknowledge that [28,29] also simulated foveation via adaptive blurring, and, point out that R-Blur builds upon these methods by (1) also simulating the loss in color sensitivity and the stochasticity of neural responses, and (2) rigorously evaluating the impact on robustness.
>
> **Were other quantified metrics [for eccentricity] used [...]?**
> We did not try other distance metrics. We do not expect a change of distance metric to significantly influence the accuracy and robustness because it will impact only a few pixels. On the other hand, their impact on the speed and memory requirements might be more significant since extracting a circular region of the image (as necessitated by the L2-norm) would be more computationally intensive than simply slicing the tensor to extract a square region of the image.
>
> We hope that our responses fully address the reviewer's concerns and we kindly request the reviewer to consider increasing their score of our paper.

---

> > ### Comment · Reviewer_ysro · 2023-08-18
> >
> > Thank you for the response. I was indeed referring to Figure 2 instead of Figure 1. Sorry, for the confusion. My concerns were appropriately addressed and I am raising my score to 7.

---

### Official Review · Reviewer_74Y4 · 2023-07-07

**Soundness:** 2 fair
**Presentation:** 3 good
**Contribution:** 3 good
**Rating:** 6
**Confidence:** 3

**Summary:**

The paper presents a biologically inspired approach that improves the robustness of DNNs against samples with adversarial perturbations or common corruptions. In the proposed approach the models are trained using the images transformed using the proposed R-Blur (Retina Blur) transformation. The proposed R-Blur simulates foveation by adaptively blurring the image pixels and reducing the color saturation based on the distance from the given fixation point. The effectiveness of the proposed approach is demonstrated by considering models trained on CIFAR-10, Ecoset, and ImageNet datasets, respectively.

**Strengths:**

The proposed biologically inspired approach is a non-adversarial training approach that yields robust models. These models are robust to unseen adversarial and common corruptions. Experimental results validate the same.

**Weaknesses:**

1. The proposed approach depends on an external model for fixation point generation. The fixation model can act as a computational and performance bottleneck. It is not clear whether the white-box evaluation considers the susceptibility of the fixation model.
2. From the ablation study, it can be seen that noise plays a critical role. The paper fails to highlight the role of noise in the proposed R-blur that simulates foveation.
3. Missing experimental details: Important experimental setup details are missing (such as architecture details, training setup, and attack setting). Furthermore, certain experiments are missing to demonstrate the effectiveness of the proposed approach (computation time, auto attack). Refer to the question section.

**Questions:**

1. Is the proposed R-blur frame differentiable?
2. For the white box adversarial attack, was the R-blur and the fixation model considered for generating the adversarial samples?
3. Is the perturbation size ($\epsilon$) defined for 0-1 or 0-255 pixel range (L178 - L179)?
4. To further validate the adversarial robustness of the proposed approach present sanity check described in [a,b].
5. How is the pixel wise fusion of color (HxWx3) and the grayscale (HxW) image performed?
6. Numbering for equation below L142 missing.
7. It is not clear why the subset of test images are used for Ecoset and ImageNet. Are the models trained on the entire training set or subset for these cases?
8. The data-augmentation techniques described in L164 to L167 are used only for the proposed approach or for all compared methods?
Sensitivity of the proposed approach to fixation method. Training details of DeepGaze-iii are missing (i) dataset and (ii) depth of the ResNet [L192].
9. Provide results for AutoAttack, apart from APGD attack.
10. In L202, L204, and L254: does "..ResNet.." mean WideResNet-22-4
11. Is the model robust to geometric attacks (e.g. translation, rotation, and affine)?
12. Provide comparison of the training and inference time for the methods considered.

[a] Carlini et al. "On Evaluating Adversarial Robustness" arxiv 2019

[b] Athalye et al. "Obfuscated Gradients Give a False Sense of Security" ICML 2018

**Limitations:**

The main limitation of the proposed approach is dependency on the external fixation model which can act as a bottleneck in terms of computation and performance. Further, it is not straightforward to adapt the proposed approach to other computer vision tasks such as segmentation, and multi-object classification. The authors are suggested to include a discussion on the same.

---

> ### Author Rebuttal · Authors · 2023-08-10
>
> We thank the reviewer for their valuable feedback and hope that our responses address their concerns
> **[...]The paper fails to highlight the role of noise [...].**
> We will highlight the role of adding noise in writing. Figure 1 in the global response shows that under moderate adversarial perturbation, the model with only Gaussian noise is almost half as accurate as a model with R-Blur on Imagenet. This result and Figure 8 in the paper show that the noise alone does not explain the robustness gains of R-Blur.
> **Is [...] R-blur [...] differentiable?**
> Yes, R-Blur is linear and fully differentiable.
>
> **[...] Were R-blur and the fixation model considered for generating the adversarial samples?**
> The white-box attack takes R-Blur into account: gradients are propagated through it to the input image. We remove the fixation model after using it to sample 5 fixation points from the clean input image, which remain constant in subsequent iterations while the attack is computed. Details in Appendix Section A.
>
> **[...] present sanity check described in [a,b].**
> We performed these checks and found no issues – more attack iterations reduce accuracy, computing the attack over the average logits from several forward passes (expectation over transformation) has no effect on accuracy, converting R-Blur to a straight-through estimator in the backward pass reduces attack effectiveness. (See Appendix Section A)
>
> **How is the pixel wise fusion [...] performed?**
>     The grayscale image is HxWx3 with the values replicated on all color channels. The pixelwise fusion is done by taking a weighted sum according to the equation on Ln142 and the following code:
>     `final=(W1*gry_blr+W2*clr_blr)/(W1+W2)`
>     Where gry_blr and clr_blr are the blurred grayscale and colored images, and W1 and W2 are HxWx1 matrices containing the gray and colored acuity estimates at each location.
>
> **[...] why the subset of test images [...]. Are the models trained on the entire training set [...]?**
> The models are trained on the entire training set. They are evaluated on the subset of the test data to speed up experimentation. We randomly shuffle the test set before extracting the subset to eliminate any biases due to dataset organization.
>
> **The data-augmentation techniques described in L164 to L167 are used only for the proposed approach or for all compared methods?**
> The same data augmentation is used to train all the models.
>
> **Sensitivity of the proposed approach to fixation method.**
> The accuracy of R-Blur on clean images _is_ sensitive to the choice of fixation point. As mentioned in Section 5, and in Appendix B, if the optimal fixation point is chosen (by exhaustive search) for each image, the accuracy of R-Blur on clean Imagenet increases from 60% to 70%, which is almost on par with the standard ResNet. Further, the robustness of R-Blur is also sensitive to the _number_ of fixation points (See  Figs 7 and 8 in the Appendix).
>
> **[...]details of DeepGaze-iii are missing [...].**
> We will update the paper with the details mentioned below:
> We used the training code from the deepgaze-iii github repo, and replaced the DenseNet-201 with the R-Warp/R-Blur augmented XResNet-18-2 trained on ImageNet. The ResNet in DeepGaze and the ResNet used for classification share the same parameters. This improves performance and reduces the additional parameters in the final model. Following [41] we train DeepGaze on SALICON (Jiang et. al, 2015). This corresponds to Phase 1 of training mentioned in Table 1 of [41]. We did not notice any benefits in our use case of the additional fine-tuning so we decided to skip Phases 2-4.
>
> M. Jiang, S. Huang, J. Duan, Q. Zhao, “SALICON: Saliency in Context”, CVPR’15
>
> **Provide results for AutoAttack [...].**
> We ran these experiments for the R-Blur augmented model trained on Imagenet and show results in Figure 1 in the global response. AutoAttack reduces the accuracy by < 3% compared to APGD, and thus would not change any of the trends observed in the paper.
>
> **Is the model robust to geometric attacks (e.g. translation, rotation, and affine)?**
> We use RandAugment and AutoAugment to introduce a lot of geometric transformations into the training data therefore _all_ the models exhibit a high degree of invariance to the geometric transformations mentioned above.
>
> **[...] comparison of the training and inference time [...].**
>
> Table 1 in the global response presents this comparison and shows that R-Blur causes minimal slowdown (1.1x compared to the vanilla ResNet) during both training and testing. Also, increasing the number of fixations slows R-Blur only sub-linearly (5 predefined fixations => 3x slowdown). Introducing dynamic fixation prediction has a greater impact on speed because each image is assigned different fixation points and so R-Blur/R-Warp can not be applied to them as a single batch. This shortcoming is likely common to most fixation transforms, and is not unique to R-Blur. In fact, under dynamic fixation prediction, R-Blur is faster than R-Warp.
>
> **[...] fixation model can act as a bottleneck [...].**
> The fixation model is not a strict dependency. As shown in the ablation study (Figure 8), removing it (and using 5 predefined fixation points) has a minor impact on the accuracy and robustness of the model. For latency-sensitive scenarios, the fixation prediction model may be removed.
>
> **[...] not straightforward to adapt [...] to other computer vision tasks [...]**
> We would appreciate it if the reviewer could elaborate on the potential hurdles they see. In most of the CV tasks a CNN is used to compute an embedding for the image. If only 1 fixation is used then the standard CNN can be swapped with a R-Blur augmented CNN. For multiple fixations, one can get the image embeddings for each fixation point independently and aggregate them by summation, concatenation, etc. We will discuss this in the paper.
>
> Some questions couldn’t be answered in the character limit but we can address them in the follow-up.

---

> > ### Comment · Reviewer_74Y4 · 2023-08-18
> >
> > Thank you for your response. Please provide the response for the following:
> >
> > 1. Regarding the adaptation of the proposed method to other CV tasks: Can the authors explain how the proposed method can be used for semantic segmentation tasks? Here, the model has to predict labels for each pixel.
> > 2. For the $l_\infty$ attacks, is the perturbation size defined for 0-1 or 0-255 pixel range (L178 - L179)?
> > 3. Provide results for the sanity check experiments described in [a,b]. Consider the CIFAR-10 dataset and $l_\infty$ attack. (i) plot of accuracy vs. perturbation size, (ii) plot of accuracy vs. attack iterations, (iii) black-box attack, and (iv) results for FGSM and PGD attacks.

---

> > > ### Author Response · Authors · 2023-08-18
> > >
> > > We thank the reviewer for responding and raising important questions. We have responded to each question below:
> > >
> > > **Q1**
> > >
> > > Several popular semantic segmentation, like DeepLab (Chen, et.al 2018) and FCN (Long, et.al 2015), extract intermediate feature maps from a pretrained deep CNN, which are then further processed by DNNs (usually CNNs) to predict logits for each semantic class at each spatial coordinate. The resulting logit map may be upscaled to match the spatial dimensions of the input image if any downsampling was involved in the earlier steps. In the case of DeepLabv3+, two feature maps are extracted from different layers of a ResNet-101. The map from the earlier layer represents low-level features, while the map from the later layer represents higher-level features. The high-level features are processed with dilated convolutions, while 1x1-convolutions are applied to low-level features. The processed low and high-level features are concatenated channel-wise and passed through a 3x3-conv to predict logits. The logit map is then upscaled to match the image size.
> > >
> > > If only 1 fixation is used, an R-Blur augmented ResNet can simply replace the vanilla CNN in most semantic segmentation models. For example, the pretrained vanilla ResNet-101 in DeepLabv3+ can simply be replaced with a pretrained R-Blur augmented ResNet-101 without the need for any further modifications. If multiple fixations are used then we would need to make some simple modifications. One option would be to run semantic segmentation and obtain logit maps independently for each fixation point and then average them to get the final logit map. Another option would be to extract low and high-level feature maps independently for each fixation point, then aggregate them by summing, averaging, or concatenation before passing them to the downstream DNN that computes logit maps from them. In either case, the modification is relatively simple and easy to implement. We expect the robustness of R-Blur augmented ResNets to carry over to any semantic segmentation model that uses them, however, verifying this is part of future work.
> > >
> > > Chen, Liang-Chieh, et al. "Encoder-decoder with atrous separable convolution for semantic image segmentation." Proceedings of the European conference on computer vision (ECCV). 2018.
> > >
> > > Long, Jonathan, Evan Shelhamer, and Trevor Darrell. "Fully convolutional networks for semantic segmentation." Proceedings of the IEEE conference on computer vision and pattern recognition. 2015.
> > >
> > > **Q2**
> > >
> > > It is defined in 0-1. The image pixels are also normalized by 255 to be in 0-1.
> > >
> > > **Q3**
> > >
> > > (i) This plot is presented in Figure 5 of the paper and the results are repeated in the following table as well. We see that accuracy decreases with increasing perturbation size.
> > >
> > > | $\|\|\epsilon\|\|_\infty$ | Accuracy |
> > > | ---------------------------- | ------------ |
> > > | 0 | 90.4 |
> > > | 0.002 | 84.3 |
> > > | 0.004 | 77.1 |
> > > | 0.008 | 55.4 |
> > >
> > > (ii) This plot for Imagenet is presented in Figure 1(a) of the appendix included with the supplementary material. Upon the reviewer's request we have repeated these experiments for CIFAR-10 using 100-step APGD with $\|\|\epsilon\|\|_\infty=0.008$ and the results are in the table below. We see that accuracy decreases with increasing number of steps.
> > >
> > > | Steps | Accuracy |
> > > | ------- | ------------- |
> > > | 1 | 61.7 |
> > > | 5 | 56.1 |
> > > | 10 | 55.6 |
> > > | 25 | 55.4 |
> > > | 100 | 55.4 |
> > >
> > > (iii) As requested by the reviewer, we evaluated the CIFAR-10 R-Blur model under the back-box Square attack [Andriushchenko et.al 2020] with $\|\|\epsilon\|\|_\infty=0.008$ and observed it achieved 64.9% accuracy. In comparison, the APGD attack was significantly more successful and brought the accuracy of the model down to 56.1% with only 5 iterations.
> > >
> > > We would also like to point out that a black-box attack is included in AutoAttack, for which we have presented results in Figure 1 of the global response.
> > >
> > > Andriushchenko, Maksym, et al. "Square attack: a query-efficient black-box adversarial attack via random search." European conference on computer vision. Cham: Springer International Publishing, 2020.
> > >
> > > (iv) Upon the reviewer's request, we evaluated R-Blur under FGSM attack and compared its accuracy with 100-step APGD in the table below. We see that for each perturbation size APGD is able to achieve lower accuracy than FGSM.
> > >
> > > | $\|\|\epsilon\|\|_\infty$ | FGSM | 100-step APGD|
> > > | ---------------------------- | -------- | ----------------- |
> > > | 0.002 | 85.2 | 84.3 |
> > > | 0.004 | 78.1 | 77.1 |
> > > | 0.008 | 61.5 | 55.4 |
> > >
> > > All these results indicate that R-Blur does not obfuscate gradients and does genuinely improve adversarial robustness. We hope we have addressed the reviewer's concerns. We would be happy to continue this discussion if further clarification is required. If no further concerns remain, we would like to request the reviewer to consider increasing the rating. Thank You.

---

> > > > ### Comment · Reviewer_74Y4 · 2023-08-19
> > > >
> > > > Thanks for the response. The major concerns has been addressed. I have updated the score.

---

### Official Review · Reviewer_6sqj · 2023-07-08

**Soundness:** 3 good
**Presentation:** 3 good
**Contribution:** 3 good
**Rating:** 5
**Confidence:** 4

**Summary:**

This paper proposes a data augmentation technique named R-Blur to improve the robustness of vision classifiers against adversarial perturbations and other non-adversarial image corruptions. The method is inspired by human visual systems where the perceived scene consists of varying levels of fidelity. As such, the training images are modified in a way that adaptive Gaussian filtering is applied centered around the fixation point in the image. Results on CIFAR-10, Ecoset, and Imagenet demonstrate that R-Blur improves robustness to adversarial perturbations and common corruptions compared to standard trained models.

**Strengths:**

Originality: There are only a few biologically inspired adversarial defense techniques. The proposed method mimics the peripheral vision in human visual systems and modifies the training images with an adaptive Gaussian filter. The approach is very unique and interesting.

Quality: The paper is well-written.

Clarity: The motivation behind the proposed method as well as the overall structure of the paper is clear. The technical details are explained clearly.

Significance: The presence of adversarial examples presents a security concern for deep neural networks utilized in various applications. This paper introduces a novel approach to bolster network robustness.

**Weaknesses:**

This paper has two main weaknesses.
1. The choice of baseline methods in evaluation. The R-Blur method, in its essence, is a Gaussian data augmentation technique, while the only non-adversarial training technique in the baseline is RandAffine. To concretely verify that the adaptive filtering from R-Blur is indeed improving the robustness of the model beyond simple Gaussian data augmentation, other baseline methods such as Gaussian augmentation (with difference variances) and l2 regularization are necessary.

2. The paper position itself as an approach to improve the adversarial robustness of deep neural networks. However, results in Sec. 3 show that the improvement in adversarial robustness towards the APGD attack is significantly lower than adversarial training. Also, the choice of $\epsilon$ in the adversarial robustness is much smaller than the standard values used in other works.

**Questions:**

Question/clarification:
Is $W_{V}$ in (1) the width of the image?
Are (2), (3), and (4) from previous work ([26]?), or are they part of the novel contributions from this paper? Why are Laplace and Cauchy distributions used? Perhaps some additional discussion following the definitions can be helpful to further improve the clarity.

In the evaluation of adversarial robustness, why is only APGD in Autoattack used, rather than the complete version of AutoAttack?

In other adversarial training methods, we can explicitly control the trade-off between standard accuracy and robust accuracy. For instance, $\beta$ in TRADES, $\epsilon$ of the perturbation in standard adversarial training. Does such a concept exist for R-Blur? I think understanding such a mechanism can further improve the adversarial robustness of models trained with R-Blur.

Suggestion:
In Figure 4, it seems that the sequence of fixation points does not converge at all. Also, from Ln182, it seems that the results are based on randomly selected fixation points. One suggestion is to identify the fixation point as the pixel location with the highest saliency. Saliency-based data augmentation (i.e., Ma et al and Uddin et al) can be a good starting point.

Be consistent with the use of gray or grey (grayscale or greyscale).


Ma, Avery, et al. "SAGE: Saliency-Guided Mixup with Optimal Rearrangements." arXiv preprint arXiv:2211.00113 (2022).
Uddin, A. F. M., et al. "Saliencymix: A saliency guided data augmentation strategy for better regularization." arXiv preprint arXiv:2006.01791 (2020).

**Limitations:**

Limitations including the trade-off between the robust accuracy and the standard accuracy are discussed.

---

> ### Author Rebuttal · Authors · 2023-08-09
>
> We hope that in our responses below we will be able to fully address the reviewer's outstanding concerns and that the reviewer will consider increasing their score of our paper.
>
> **verify that the adaptive filtering from R-Blur is indeed improving the robustness of the model beyond simple Gaussian data augmentation**
>
> To verify if adaptive blurring in R-Blur does indeed contribute to robustness we compare R-Blur with models that either add Gaussian noise (with the same variance as the noise in R-Blur) or perform non-adaptive Gaussian blurring (with the variance equal to the maximum variance used by R-Blur). The results are presented in Figure 8 and we observe that both of these models are less robust than R-Blur. We have repeated this analysis for Imagenet models ( Figure 1 of the global response) and observed that the R-Blur model consistently achieves higher accuracy than baseline methods on all adversarial perturbation sizes.
>
> We also trained models augmented with non-adaptive Gaussian blur of different variances, as well as a model that combines non-adaptive Gaussian blur with Gaussian noise (Figure 4 of the global response), and found R-Blur to have superior robustness then all of them
>
> ** improvement in adversarial robustness towards the APGD attack is significantly lower than adversarial training.**
>
> We would like to clarify that the objective of this paper was to study the impact of foveation on the robustness of DNNs and we have not claimed that R-Blur is superior to adversarial training as a defense. Our claims are that (1) R-Blur improves the robustness of CNNs to image perturbation – both adversarial and non-adversarial, compared to the vanilla CNN and other biologically-inspired methods, and (2) unlike adversarial training, which seems to have negligible effect on robustness to non-adversarial image perturbations, R-Blur increases robustness to a variety of image perturbations, not just L_p bounded adversarial attacks. We believe that these claims are justified by the results. Nevertheless, we will rewrite certain parts of the paper to ensure that no confusion remains regarding our objectives and claims.
>
> **the choice of eps in the adversarial robustness**
>
> We sought to validate the claim that R-Blur improves robustness to perturbations – both adversarial and non-adversarial, compared to the vanilla CNN and other biologically-inspired methods. Therefore we chose $\epsilon$ values on which the compared models had accuracy greater than 0%. As shown in Figure 5 in the paper and Figure 1 in the global response, the accuracy of all the models, except AT, on ecoset/imagenet becomes close to 0% on the largest perturbation we used so evaluating at larger perturbation would not have served any purpose. Furthermore, the perturbation sizes ($\epsilon$) used in our evaluation are on par or higher than the sizes used in papers on biologically-plausible adversarial defenses [18,21,22].
>
> **Is W_V in (1) the width of the image?**
>
> Yes, W_V is the width of the image – 224.
>
> **Are (2), (3), and (4) from previous work ([26]?), [...]?**
>
> The equations 2, 3, and 4 are not from previous work [26]. We devised these equations to match the curves of visual acuity presented in Figure 14.21 (B) in [26].
>
> **Why are Laplace and Cauchy distributions used?**
>
> The Laplace and Cauchy distributions are chosen to approximate the shape of the curves in Figure 14.21 (B). We will update the paper to make this clearer.
>
> **why is only APGD in Autoattack used**
>
> We used only APGD because (1) the certified accuracy of R-Blur (Figure 7) was similar to empirically measured accuracy under adversarial perturbations (Figure 5 top row), and (2) our analysis in Section A of the Appendix revealed no gradient obfuscation effects. Under these conditions, we were confident that evaluating with only APGD would give a reliable measure of robustness.
>
> At the reviewers’ request, we ran these experiments for the R-Blur augmented model trained on Imagenet. Figure 3 in the global response shows that the accuracy under AutoAttack is only slightly lower than the accuracy under APGD, with the maximum difference being 3%, which would not change any of the trends observed in the paper.
>
> **Can the accuracy-robustness tradeoff be controlled for R-Blur?**
>
> The accuracy and robustness can be traded with each other by controlling the noise added in R-Blur. Figure 6 in the Appendix illustrates this trade-off. We see that as the variance of the noise is increased from 0.125 to 0.5 the clean accuracy drops from 90% to 85%, while the accuracy under attack increases from around 30% to 50%.
>
> **In Figure 4, [...] the sequence of fixation points does not converge**
>
> The fixation points in Figure 4 do not converge to a single spatial location by design, as stated in the caption. This is in line with human/animal behavior, where it has been observed that they fixate on different salient parts of the scene to progressively accumulate more information.
>
> **from Ln182, it seems that the results are based on randomly selected fixation points.**
>
> We would like to point out that the Ln180-182 talk about the training set up. However, as mentioned in Ln185-185, during inference a model of human gaze (DeepGaze-III) is used to select the fixation points.
>
> **One suggestion is to identify the fixation point as the pixel location with the highest saliency.**
>
> While the reviewer’s suggestion is well taken and we thank the reviewer for sharing relevant papers, we would like to clarify that our approach does in fact subsume this technique. We use DeepGaze-III to predict a saliency map for a given image (see Figure 4 top-row). From this saliency map we pick the most salient coordinate as the fixation point. We repeat this process, as shown in Figure 4, to get a sequence of multiple fixation points. We will rewrite the relevant parts of the paper to ensure that it is clear that we are already using saliency maps to select fixation points.

---

> > ### Comment · Reviewer_6sqj · 2023-08-17
> >
> > Thank you for the clarification. Most of my concerns have been addressed. I will raise my score to 5.

---

> > > ### Author Response · Authors · 2023-08-19
> > >
> > > We are glad that we were able to fully address the reviewer's concerns and we thank the reviewer for increasing their score. Given that 5 is a borderline score, we wanted to ask if there are any lingering questions or concerns due to which the reviewer is currently unable to give a higher score to our paper? We are committed to working with the reviewer to address any issues and improve our paper.

---

> > > > ### Comment · Reviewer_6sqj · 2023-08-19
> > > >
> > > > The main reason for the score comes from the limited improvement in adversarial robustness.

---

> > > > > ### Author Response · Authors · 2023-08-19
> > > > >
> > > > > We thank the reviewer for responding. We understand the reviewer's reservation that the improvement in adversarial robustness is not very large, and we respect the reviewer's decision regarding the score.
> > > > >
> > > > > We would like to point out that the adversarial robustness of R-Blur does not arise from being trained on adversarially perturbed data, so the gains in adversarial robustness are expected to be lower than adversarial training. This is because the gap between the training and testing distribution for adversarial training is much smaller than it is for R-Blur. Furthermore, unlike adversarial training, the robustness of R-Blur is not limited to norm-bounded adversarial attacks and extends to non-adversarial corruptions. On the other hand, an adversarial trained model is about as robust to non-adversarial corruptions as a vanilla resnet. Finally, the adversarial and non-adversarial robustness of R-Blur is significantly better than baseline models and contemporary biologically motivated methods.

---

### Author Rebuttal · Authors · 2023-08-09

We are very thankful to the reviewers for taking the time to read our paper carefully and providing valuable feedback that will undoubtedly help strengthen the paper and increase its impact. We also thank the reviewers for asking thoughtful questions and raising important concerns.

We have responded to each reviewer’s questions and concerns in separate responses to their respective reviews. Due to space constraints we were unable to respond to comments about editorial issues, like formatting, typos and missing citations. Nevertheless, we would like to assure the reviewers that we acknowledge those comments and we will update the paper accordingly for the camera ready. To reduce ambiguity we answer each question/concern separately, quoting the reviewers' words (in bold font), either verbatim or summarized, before providing our response. If there are instances where our responses do not fully address the reviewers’ concerns or if it appears that we may have misunderstood the reviewers’ intent, we encourage the reviewers to ask follow-up questions and allow us the opportunity better understand their point of view and to provide additional details and clarification. If our responses do adequately address the reviewers’ concerns we request the reviewers to consider increasing their scores.

We have also performed some additional experiments and analyses based on requests by the reviewers and have presented the results in the PDF attached to this response. The tables and figures in the PDF are referenced in the responses to the specific questions/comments that requested them. The list of tables and figures in the PDF is also listed below:

-   Table 1: Computation time of all the methods compared in this paper

-   Figure 1: Accuracy of R-Blur, baseline methods, adversarial training, non-adaptive Gaussian blur, and Gaussian noise at different levels of adversarial perturbation.

-   Figure 2: A corrected version of Figure 2 from the paper.

-   Figure 3: Accuracy of R-Blur under APGD and AutoAttack at different levels of adversarial perturbation.

-   Figure 4: Comparison of accuracy under clear images and adversarial attacks between R-Blur and models with Gaussian blurring of difference variances.

Notes:
-  We refer to the Appendix in our responses below. This appendix is included in the supplementary material submitted with the paper.

---

### Comment · Area_Chair_E8eG · 2023-08-16
**Discussion**

Dear Reviewers,

so far, there has not been any discussion on this paper with the authors. In case you have any further questions after reading the rebuttal, please use the opportunity of the discussion phase to ask the authors!

Best regards,
AC

---

### Decision · Program_Chairs · 2023-09-21

**Decision:**

Accept (poster)

**Comment:**

The paper proposes a biologically inspired approach to improve model robustness towards adversarial perturbations and common corruptions. It leverages fixation points as basis for data augmentation during training. Specifically, it mimics retina blur (thus the name R-Blur) and reduces the color saturation depending on the distance from the fixation point to simulate a foveated view on the respective images. Evaluations are provided on CIFAR-10, Ecoset, and ImageNet and show the benefit of the proposed approach.
The paper is well written and provides a well-motivated link between model robustness and human perception. While the reviewers had initial concerns on the validity of baselines (e.g. including Gaussian blurring and the number of augmented samples), these concerns could be clarified during the rebuttal and discussion phase. After the discussion, the paper has a clear accept vote from reviewers and the area chair. The authors are encouraged to include the additional results from the rebuttal phase as well as the discussion on the role of noise into their final submission.